# Online Non-Convex Optimization
# with Imperfect Feedback

**Amélie Héliou**
Criteo AI Lab
a.heliou@criteo.com

**Matthieu Martin**
Criteo AI Lab
mat.martin@criteo.com

**Panayotis Mertikopoulos**
Univ. Grenoble Alpes, CNRS, Inria, LIG &
Criteo AI Lab
panayotis.mertikopoulos@imag.fr

**Thibaud Rahier**
Criteo AI Lab
t.rahier@criteo.com

## Abstract

We consider the problem of online learning with non-convex losses. In terms of feedback, we assume that the learner observes – or otherwise constructs – an inexact model for the loss function encountered at each stage, and we propose a mixed-strategy learning policy based on dual averaging. In this general context, we derive a series of tight regret minimization guarantees, both for the learner's static (external) regret, as well as the regret incurred against the best *dynamic* policy in hindsight. Subsequently, we apply this general template to the case where the learner only has access to the actual loss incurred at each stage of the process. This is achieved by means of a kernel-based estimator which generates an inexact model for each round's loss function using only the learner's realized losses as input.

## 1   Introduction

In this paper, we consider the following online learning framework:

1. At each stage $t = 1, 2, \dots$ of a repeated decision process, the learner selects an action $x_t$ from a compact convex subset $\mathcal{K}$ of a Euclidean space $\mathbb{R}^n$.
2. The agent's choice of action triggers a loss $\ell_t(x_t)$ based on an a priori unknown *loss function* $\ell_t \colon \mathcal{K} \to \mathbb{R}$; subsequently, the process repeats.

If the loss functions $\ell_t$ encountered by the agent are *convex*, the above framework is the standard online convex optimization setting of Zinkevich [58] – for a survey, see [16, 29, 45] and references therein. In this case, simple first-order methods like online gradient descent (OGD) allow the learner to achieve $\mathcal{O}(T^{1/2})$ regret after $T$ rounds [58], a bound which is well-known to be min-max optimal in this setting [1, 45]. At the same time, it is also possible to achieve tight regret minimization guarantees against *dynamic comparators* – such as the regret incurred against the best *dynamic policy* in hindsight, cf. [11, 21, 23, 30, 33] and references therein.

On the other hand, when the problem's loss functions are not convex, the situation is considerably more difficult. When the losses are generated from a stationary stochastic distribution, the problem can be seen as a version of a continuous-armed bandit in the spirit of Agrawal [3]; in this case, there exist efficient algorithms guaranteeing logarithmic regret by discretizing the problem's search domain and using a UCB-type policy [18, 37, 48]. Otherwise, in an adversarial context, an informed adversary can impose *linear* regret to *any deterministic algorithm* employed by the learner [31, 45, 50]; as a result, UCB-type approaches are no longer suitable.

| Feedback | Convex losses | | Non-Convex losses | |
|---|---|---|---|---|
| | Static regret | Dynamic regret | Static regret | Dynamic regret |
| Exact | $\mathcal{O}(T^{1/2})$ [58] | $\mathcal{O}(T^{2/3}V_T^{1/3})$ [11] | $\mathcal{O}(T^{1/2})$ [38, 50] | $\mathcal{O}(T^{2/3}V_T^{1/3})$ |
| Unbiased | $\mathcal{O}(T^{1/2})$ [58] | $\mathcal{O}(T^{2/3}V_T^{1/3})$ [11] | $\mathbf{\mathcal{O}(T^{1/2})}$ | $\mathbf{\mathcal{O}(T^{2/3}V_T^{1/3})}$ |
| Bandit | $\mathcal{O}(T^{1/2})$ [17, 19] | $\mathcal{O}(T^{4/5}V_T^{1/5})$ [11] | $\mathbf{\mathcal{O}(T^{\frac{n+2}{n+3}})}$ | $\mathbf{\mathcal{O}(T^{\frac{n+3}{n+4}}V_T^{\frac{1}{n+4}})}$ |

**Table 1:** Overview of related work. In regards to feedback, an "exact" model means that the learner acquires perfect knowledge of the encountered loss functions; "unbiased" refers to an inexact model that is only accurate on average; finally, "bandit" means that the learner records their incurred loss and has no other information. We only report here the best known bounds in the literature; all bounds derived in this paper are typeset **in bold**.

In view of this impossibility result, two distinct threads of literature have emerged for online non-convex optimization. One possibility is to examine less demanding measures of regret – like the learner's *local regret* [31] – and focus on first-order methods that minimize it efficiently [28, 31]. Another possibility is to consider *randomized* algorithms, in which case achieving no regret *is* possible: Krichene et al. [38] showed that adapting the well-known *Hedge* (or multiplicative / exponential weights) algorithm to a continuum allows the learner to achieve $\mathcal{O}(T^{1/2})$ regret, as in the convex case. This result is echoed in more recent works by Agarwal et al. [2] and Suggala & Netrapalli [50] who analyzed the *"follow the perturbed leader"* (FTPL) algorithm of Kalai & Vempala [34] with exponentially distributed perturbations and an offline optimization oracle (exact or approximate); again, the regret achieved by FTPL in this setting is $\mathcal{O}(T^{1/2})$, i.e., order-equivalent to that of Hedge in a continuum.

**Our contributions and related work.** A crucial assumption in the above works on randomized algorithms is that, after selecting an action, the learner receives perfect information on the loss function encountered – i.e., an *exact model* thereof. This is an important limitation for the applicability of these methods, which led to the following question by Krichene et al. [38, p. 8]:

> *One question is whether one can generalize the Hedge algorithm to a bandit setting, so that sublinear regret can be achieved without the need to explicitly maintain a cover.*

To address this open question, we begin by considering a general framework for randomized action selection with *imperfect feedback* – i.e., with an inexact model of the loss functions encountered at each stage. Our contributions in this regard are as follows:

1. We present a flexible algorithmic template for online non-convex learning based on dual averaging with imperfect feedback [42].

2. We provide tight regret minimization rates – both *static* and *dynamic* – under a wide range of different assumptions for the loss models available to the optimizer.

3. We show how this framework can be extended to learning with *bandit feedback*, i.e., when the learner only observes their realized loss and must construct a loss model from scratch.

Viewed abstractly, the dual averaging (DA) algorithm is an "umbrella" scheme that contains Hedge as a special case for problems with a simplex-like domain. In the context of online convex optimization, the method is closely related to the well-known "follow the regularized leader" (FTRL) algorithm of Shalev-Shwartz & Singer [46], the FTPL method of Kalai & Vempala [34], "lazy" mirror descent (MD) [15, 16, 45], etc. For an appetizer to the vast literature surrounding these methods, we refer the reader to [14, 16, 42, 45, 46, 55, 57] and references therein.

In the *non-convex* setting, our regret minimization guarantees can be summarized as follows (see also Table 1 above): if the learner has access to inexact loss models that are unbiased and finite in mean square, the DA algorithm achieves in expectation a *static* regret bound of $\mathcal{O}(T^{1/2})$. Moreover, in terms of the learner's *dynamic* regret, the algorithm enjoys a bound of $\mathcal{O}(T^{2/3}V_T^{1/3})$ where $V_T := \sum_{t=1}^{T}\|\ell_{t+1} - \ell_t\|_\infty$ denotes the *variation* of the loss functions encountered over the horizon of play (cf. Section 4 for the details). Importantly, both bounds are order-optimal, even in the context of online *convex* optimization, cf. [1, 11, 20].

With these general guarantees in hand, we tackle the bandit setting using a "kernel smoothing" technique in the spirit of Bubeck et al. [19]. This leads to a new algorithm, which we call *bandit dual averaging* (BDA), and which can be seen as a version of the DA method with *biased* loss models. The bias of the loss model can be controlled by tuning the "radius" of the smoothing kernel; however, this comes at the cost of increasing the model's variance – an incarnation of the well-known "bias-variance" trade-off. By resolving this trade-off, we are finally able to answer the question of Krichene et al. [38] in the positive: BDA enjoys an $\mathcal{O}(T^{\frac{n+2}{n+3}})$ static regret bound and an $\mathcal{O}(T^{\frac{n+3}{n+4}}V_T^{1/(n+4)})$ *dynamic* regret bound, without requiring an explicit discretization of the problem's search space.

This should be contrasted with the case of online *convex* learning, where it is possible to achieve $\mathcal{O}(T^{3/4})$ regret through the use of simultaneous perturbation stochastic approximation (SPSA) techniques [25], or even $\mathcal{O}(T^{1/2})$ by means of kernel-based methods [17, 19]. This represents a drastic drop from $\mathcal{O}(T^{1/2})$, but this cannot be avoided: the worst-case bound for stochastic non-convex optimization is $\Omega(T^{(n+1)/(n+2)})$ [36, 37], so our static regret bound is nearly optimal in this regard (i.e., up to $\mathcal{O}(T^{-1/(n+2)(n+3)})$, a term which is insignificant for horizons $T \leq 10^{12}$). Correspondingly, in the case of dynamic regret minimization, the best known upper bound is $\mathcal{O}(T^{4/5}V_T^{1/5})$ for online *convex* problems [11, 24]. We are likewise not aware of any comparable dynamic regret bounds for online *non-convex* problems; to the best our knowledge, our paper is the first to derive dynamic regret guarantees for online non-convex learning with bandit feedback.

We should stress here that, as is often the case for methods based on lifting, much of the computational cost is hidden in the sampling step. This is also the case for the proposed DA method which, like [38], implicitly assumes access to a sampling oracle. Estimating (and minimizing) the per-iteration cost of sampling is an important research direction, but one that lies beyond the scope of the current paper, so we do not address it here.

## 2  Setup and preliminaries

**2.1. The model.**   Throughout the sequel, our only blanket assumption will be as follows:

**Assumption 1.**  The stream of loss functions encountered is *uniformly bounded Lipschitz*, i.e., there exist constants $R, L > 0$ such that, for all $t = 1, 2, \ldots$, we have:

1. $|\ell_t(x)| \leq R$ for all $x \in \mathcal{K}$; more succinctly, $\|\ell_t\|_\infty \leq R$.
2. $|\ell_t(x') - \ell_t(x)| \leq L\|x' - x\|$ for all $x, x' \in \mathcal{K}$.

Other than this meager regularity requirement, we make no structural assumptions for $\ell_t$ (such as convexity, unimodality, or otherwise). In this light, the framework under consideration is akin to the online non-convex setting of Krichene et al. [38], Hazan et al. [31], and Suggala & Netrapalli [50]. The main difference with the setting of Krichene et al. [38] is that the problem's domain $\mathcal{K}$ is assumed convex; this is done for convenience only, to avoid technical subtleties involving "uniform fatness" conditions and the like.

In terms of playing the game, we will assume that the learner can employ *mixed strategies* to randomize their choice of action at each stage; however, because this mixing occurs over a *continuous* domain, defining this randomization requires some care. To that end, let $\mathcal{M} \equiv \mathcal{M}(\mathcal{K})$ denote the space of all finite signed Radon measures on $\mathcal{K}$. Then, a *mixed strategy* is defined as an element $\pi$ of the set of Radon probability measures $\Delta \equiv \Delta(\mathcal{K}) \subseteq \mathcal{M}(\mathcal{K})$ on $\mathcal{K}$, and the player's expected loss under $\pi$ when facing a bounded loss function $\ell \in \mathcal{L}^\infty(\mathcal{K})$ will be denoted as

$$\langle \ell, \pi \rangle \coloneqq \mathbb{E}_\pi[\ell] = \int_\mathcal{K} \ell(x)\, d\pi(x). \tag{1}$$

*Remark* 1.  We should note here that $\Delta$ contains a vast array of strategies, including atomic and singular distributions that do not admit a density. For this reason, we will write $\Delta_c$ for the set of strategies that are *absolutely continuous* relative to the Lebesgue measure $\lambda$ on $\mathcal{K}$, and $\Delta_\perp$ for the set of singular strategies (which are not); by Lebesgue's decomposition theorem [26], we have $\Delta = \Delta_c \cup \Delta_\perp$. By construction, $\Delta_\perp$ contains the player's *pure strategies*, i.e., Dirac point masses $\delta_x$ that select $x \in \mathcal{K}$ with probability 1; however, it also contains pathological strategies that admit *neither* a density, *nor* a point mass function – such as the Cantor distribution [26]. By contrast, the Radon-Nikodym (RN) derivative $p \coloneqq d\pi/d\lambda$ of $\pi$ exists for all $\pi \in \Delta_c$, so we will sometimes refer

to elements of $\Delta_c$ as "Radon-Nikodym strategies"; in particular, if $\pi \in \Delta_c$, we will not distinguish between $\pi$ and $p$ unless absolutely necessary to avoid confusion.

Much of our analysis will focus on strategies $\chi$ with a piecewise constant density on $\mathcal{K}$, i.e., $\chi = \sum_{i=1}^{k} \alpha_i \mathbb{1}_{\mathcal{C}_i}$ for a collection of weights $\alpha_i \geq 0$ and measurable subsets $\mathcal{C}_i \subseteq \mathcal{K}$, $i = 1, \ldots, k$, such that $\int_{\mathcal{K}} \chi = \sum_i \alpha_i \lambda(\mathcal{C}_i) = 1$. These strategies will be called *simple* and the space of simple strategies on $\mathcal{K}$ will be denoted by $\mathcal{X} \equiv \mathcal{X}(\mathcal{K})$. A key fact regarding simple strategies is that $\mathcal{X}$ is dense in $\Delta$ in the weak topology of $\mathcal{M}$ [26, Chap. 3]; as a result, the learner's expected loss under *any* mixed strategy $\pi \in \Delta$ can be approximated within arbitrary accuracy $\varepsilon > 0$ by a simple strategy $\chi \in \mathcal{X}$. In addition, when $k$ (or $n$) is not too large, sampling from simple strategies can be done efficiently; for all these reasons, simple strategies will play a key role in the sequel.

**2.2. Measures of regret.** With all this in hand, the *regret* of a learning policy $\pi_t \in \Delta$, $t = 1, 2, \ldots$, against a benchmark strategy $\pi^* \in \Delta$ is defined as

$$\mathrm{Reg}_{\pi^*}(T) = \sum_{t=1}^{T} \big[ \mathbb{E}_{\pi_t}[\ell_t] - \mathbb{E}_{\pi^*}[\ell_t] \big] = \sum_{t=1}^{T} \langle \ell_t, \pi_t - \pi^* \rangle, \tag{2}$$

i.e., as the difference between the player's mean cumulative loss under $\pi_t$ and $\pi^*$ over $T$ rounds. In a slight abuse of notation, we write $\mathrm{Reg}_{p^*}(T)$ if $\pi^*$ admits a density $p^*$, and $\mathrm{Reg}_x(T)$ for the regret incurred against the pure strategy $\delta_x$, $x \in \mathcal{K}$. Then, the player's (*static*) *regret* under $\pi_t$ is given by

$$\mathrm{Reg}(T) = \max_{x \in \mathcal{K}} \mathrm{Reg}_x(T) = \sup_{\chi \in \mathcal{X}} \mathrm{Reg}_\chi(T) \tag{3}$$

where the maximum is justified by the compactness of $\mathcal{K}$ and the continuity of each $\ell_t$. The lemma below provides a link between pure comparators and their approximants in the spirit of Krichene et al. [38]; to streamline our discussion, we defer the proof to the supplement:

**Lemma 1.** *Let $\mathcal{U}$ be a convex neighborhood of $x$ in $\mathcal{K}$ and let $\chi \in \mathcal{X}$ be a simple strategy supported on $\mathcal{U}$. Then, $\mathrm{Reg}_x(T) \leq \mathrm{Reg}_\chi(T) + L \operatorname{diam}(\mathcal{U})T$.*

This lemma will be used to bound the agent's static regret using bounds obtained for simple strategies $\chi \in \mathcal{X}$. Going beyond static comparisons of this sort, the learner's *dynamic regret* is defined as

$$\mathrm{DynReg}(T) = \sum_{t=1}^{T} [\langle \ell_t, \pi_t \rangle - \min_{\pi \in \Delta} \langle \ell_t, \pi \rangle] = \sum_{t=1}^{T} \langle \ell_t, \pi_t - \pi_t^* \rangle \tag{4}$$

where $\pi_t^* \in \arg\min_{\pi \in \Delta} \langle \ell_t, \pi \rangle$ is a "best-response" to $\ell_t$ (that such a strategy exists is a consequence of the compactness of $\mathcal{K}$ and the continuity of each $\ell_t$). In regard to its static counterpart, the agent's dynamic regret is considerably more ambitious, and achieving sublinear dynamic regret is not always possible; we examine this issue in detail in Section 4.

**2.3. Feedback models.** After choosing an action, the agent is only assumed to observe an *inexact model* $\hat{\ell}_t \in \mathcal{L}^\infty(\mathcal{K})$ of the $t$-th stage loss function $\ell_t$; for concreteness, we will write

$$\hat{\ell}_t = \ell_t + e_t \tag{5}$$

where the "observation error" $e_t$ captures all sources of uncertainty in the player's model. This uncertainty could be both "random" (zero-mean) or "systematic" (non-zero-mean), so it will be convenient to decompose $e_t$ as

$$e_t = z_t + b_t \tag{6}$$

where $z_t$ is zero-mean and $b_t$ denotes the mean of $e_t$.

To define all this formally, we will write $\mathcal{F}_t = \mathcal{F}(\pi_1, \ldots, \pi_t)$ for the history of the player's mixed strategy up to stage $t$ (inclusive). The chosen action $x_t$ and the observed model $\hat{\ell}_t$ are both generated *after* the player chooses $\pi_t$ so, by default, they are not $\mathcal{F}_t$-measurable. Accordingly, we will collect all randomness affecting $\hat{\ell}_t$ in an abstract probability law $\mathbb{P}$, and we will write $b_t = \mathbb{E}[e_t \mid \mathcal{F}_t]$ and $z_t = e_t - b_t$; in this way, $\mathbb{E}[z_t \mid \mathcal{F}_t] = 0$ by definition.

In view of all this, we will focus on the following descriptors for $\hat{\ell}_t$:

    a)  *Bias:* $\qquad\qquad\qquad \|b_t\|_\infty \leq B_t$                                                             (7a)

    b)  *Variance:* $\qquad\quad \mathbb{E}[\|z_t\|_\infty^2 \mid \mathcal{F}_t] \leq \sigma_t^2$                                    (7b)

    c)  *Mean square:* $\quad\;\; \mathbb{E}[\|\hat{\ell}_t\|_\infty^2 \mid \mathcal{F}_t] \leq M_t^2$                                 (7c)

In the above, $B_t$, $\sigma_t$ and $M_t$ are deterministic constants that are to be construed as bounds on the bias, (conditional) variance, and magnitude of the model $\hat{\ell}_t$ at time $t$. In obvious terminology, a model with $B_t = 0$ will be called *unbiased*, and an unbiased model with $\sigma_t = 0$ will be called *exact*.

▶ **Example 1** (Parametric models). An important application of online optimization is the case where the encountered loss functions are of the form $\ell_t(x) = \ell(x; \theta_t)$ for some sequence of parameter vectors $\theta_t \in \mathbb{R}^m$. In this case, the learner typically observes an estimate $\hat{\theta}_t$ of $\theta_t$, leading to the inexact model $\hat{\ell}_t = \ell(\cdot; \hat{\theta}_t)$. Importantly, this means that $\hat{\ell}_t$ *does not require infinite-dimensional feedback* to be constructed. Moreover, the dependence of $\ell$ on $\theta$ is often linear, so if $\hat{\theta}_t$ is an unbiased estimate of $\theta_t$, then so is $\hat{\ell}_t$. ◀

▶ **Example 2** (Online clique prediction). As a specific incarnation of a parametric model, consider the problem of finding the largest complete subgraph – a *maximum clique* – of an undirected graph $\mathcal{G} = (\mathcal{V}, \mathcal{E})$. This is a key problem in machine learning with applications to social networks [27], data mining [9], gene clustering [49], feature embedding [56], and many other fields. In the online version of the problem, the learner is asked to predict such a clique in a graph $\mathcal{G}_t$ that evolves over time (e.g., a social network), based on partial historical observations of the graph. Then, by the Motzkin-Straus theorem [13, 41], this boils down to an online quadratic program of the form:

$$\text{maximize } u_t(x) = \sum_{i,j=1}^n x_i A_{ij,t} x_j \qquad \text{subject to } x_i \geq 0, \ \sum_{i=1}^n x_i = 1, \qquad \text{(MCP)}$$

where $A_t = (A_{ij,t})_{i,j=1}^n$ denotes the adjacency matrix of $\mathcal{G}_t$. Typically, $\hat{A}_t$ is constructed by picking a node $i$ uniformly at random, charting out its neighbors, and letting $\hat{A}_{ij,t} = |\mathcal{V}|/2$ whenever $j$ is connected to $i$. It is easy to check that $\hat{A}_t$ is an unbiased estimator of $A_t$; as a result, the function $\hat{u}_t(x) = x^\top \hat{A}_t x$ is an unbiased model of $u_t$. ◀

▶ **Example 3** (Online-to-batch). Consider an empirical risk minimization model of the form

$$f(x) = \tfrac{1}{m} \sum_{i=1}^m f_i(x) \tag{8}$$

where each $f_i \colon \mathbb{R}^n \to \mathbb{R}$ corresponds to a data point (or "sample"). In the "online-to-batch" formulation of the problem [45], the optimizer draws uniformly at random a sample $i_t \in \{1, \dots, m\}$ at each stage $t = 1, 2, \dots$, and observes $\hat{\ell}_t = f_{i_t}$. Typically, each $f_i$ is relatively easy to store in closed form, so $\hat{\ell}_t$ is an easily available unbiased model of the empirical risk function $f$. ◀

## 3 Prox-strategies and dual averaging

The class of non-convex online learning policies that we will consider is based on the general template of *dual averaging* (DA) / *"follow the regularized leader"* (FTRL) methods. Informally, this scheme can be described as follows: at each stage $t = 1, 2, \dots$, the learner plays a mixed strategy that minimizes their cumulative loss up to round $t - 1$ (inclusive) plus a "regularization" penalty term (hence the "regularized leader" terminology). In the rest of this section, we provide a detailed construction and description of the method.

**3.1. Randomizing over discrete vs. continuous sets.** We begin by describing the dual averaging method when the underlying action set is *finite*, i.e., of the form $\mathcal{A} = \{1, \dots, n\}$. In this case, the space of mixed strategies is the $n$-dimensional simplex $\Delta_n := \Delta(\mathcal{A}) = \{p \in \mathbb{R}_+^n : \sum_{i=1}^n p_i = 1\}$, and, at each $t = 1, 2, \dots$, the dual averaging algorithm prescribes the mixed strategy

$$p_t \leftarrow \arg\min_{p \in \Delta_n} \{\eta \sum_{s=1}^{t-1} \langle \hat{\ell}_t, p \rangle + h(p)\}. \tag{9}$$

In the above, $\eta > 0$ is a "learning rate" parameter and $h \colon \Delta_n \to \mathbb{R}$ is the method's "regularizer", assumed to be continuous and strongly convex over $\Delta_n$. In this way, the algorithm can be seen as tracking the "best" choice up to the present, modulo a "day 0" regularization component – the "follow the regularized leader" interpretation.

In our case however, the method is to be applied to the *infinite-dimensional* set $\Delta \equiv \Delta(\mathcal{K})$ of the learner's mixed strategies, so the issue becomes considerably more involved. To illustrate the problem, consider one of the prototypical regularizer functions, the negentropy $h(p) = \sum_{i=1}^n p_i \log p_i$ on $\Delta_n$. If we naïvely try to extend this definition to the infinite-dimensional space $\Delta(\mathcal{K})$, we immediately run into problems: First, for pure strategies, any expression of the form $\sum_{x' \in \mathcal{K}} \delta_x(x') \log \delta_x(x')$ would be meaningless. Second, even if we focus on Radon-Nikodym strategies $p \in \Delta_c$ and use the integral definition $h(p) = \int_\mathcal{K} p \log p$, a density like $p(x) \propto 1/(x(\log x)^2)$ on $\mathcal{K} = [0, 1/2]$ has *infinite* negentropy, implying that even $\Delta_c$ is too large to serve as a domain.

**3.2. Formal construction of the algorithm.** To overcome the issues identified above, our starting point will be that any mixed-strategy incarnation of the dual averaging algorithm must contain *at least* the space $\mathcal{X} \equiv \mathcal{X}(\mathcal{K})$ of the player's simple strategies. To that end, let $\mathcal{V}$ be an ambient Banach space which contains the set of simple strategies $\mathcal{X}$ as an embedded subset. For technical reasons, we will also assume that the topology induced on $\mathcal{X}$ by the reference norm $\|\cdot\|$ of $\mathcal{V}$ is not weaker than the natural topology on $\mathcal{X}$ induced by the total variation norm; formally, $\|\cdot\|_{\mathrm{TV}} \leq \alpha\|\cdot\|$ for some $\alpha > 0$.[1] For example, $\mathcal{V}$ could be the (Banach) space $\mathcal{M}(\mathcal{K})$ of finite signed measures on $\mathcal{K}$, the (Hilbert) space $\mathcal{L}^2(\mathcal{K})$ of square integrable functions on $\mathcal{K}$ endowed with the $L^2$ norm,[2] etc.

With all this in hand, the notion of a regularizer on $\mathcal{X}$ is defined as follows:

**Definition 1.** A *regularizer* on $\mathcal{X}$ is a lower semi-continuous (l.s.c.) convex function $h\colon \mathcal{V} \to \mathbb{R} \cup \{\infty\}$ such that:

1. $\mathcal{X}$ is a weakly dense subset of the *effective domain* $\operatorname{dom} h \coloneqq \{p : h(p) < \infty\}$ of $h$.

2. The subdifferential $\partial h$ of $h$ admits a *continuous selection*, i.e., there exists a continuous mapping $\nabla h$ on $\operatorname{dom} \partial h \coloneqq \{\partial h \neq \varnothing\}$ such that $\nabla h(q) \in \partial h(q)$ for all $q \in \operatorname{dom} \partial h$.

3. $h$ is strongly convex, i.e., there exists some $K > 0$ such that $h(p) \geq h(q) + \langle \nabla h(q), p - q \rangle + (K/2)\|p - q\|^2$ for all $p \in \operatorname{dom} h$, $q \in \operatorname{dom} \partial h$.

The set $\mathcal{Q} \coloneqq \operatorname{dom} \partial h$ will be called the *prox-domain* of $h$; its elements will be called *prox-strategies*.

*Remark.* For completeness, recall that the subdifferential of $h$ at $q$ is the set $\partial h(q) = \{\psi \in \mathcal{V}^* : h(p) \geq h(q) + \langle \psi, p - q \rangle \text{ for all } q \in \mathcal{V}\}$; also lower semicontinuity means that the sublevel sets $\{h \leq c\}$ of $h$ are closed for all $c \in \mathbb{R}$. For more details, we refer the reader to Phelps [44].

Some prototypical examples of this general framework are as follows (with more in the supplement):

▶ **Example 4** ($L^2$ regularization). Let $\mathcal{V} = \mathcal{L}^2(\mathcal{K})$ and consider the quadratic regularizer $h(p) = (1/2)\|p\|_2^2 = (1/2)\int_{\mathcal{K}} p^2$ if $p \in \Delta_c \cap \mathcal{L}^2(\mathcal{K})$, and $h(p) = \infty$ otherwise. In this case, $\mathcal{Q} = \operatorname{dom} h = \Delta_c \cap \mathcal{L}^2(\mathcal{K})$ and $\nabla h(q) = q$ is a continuous selection of $\partial h$ on $\mathcal{Q}$. ◀

▶ **Example 5** (Entropic regularization). Let $\mathcal{V} = \mathcal{M}(\mathcal{K})$ and consider the entropic regularizer $h(p) = \int_{\mathcal{K}} p \log p$ whenever $p$ is a density with finite entropy, $h(p) = \infty$ otherwise. By Pinsker's inequality, $h$ is 1-strongly convex relative to the total variation norm $\|\cdot\|_{\mathrm{TV}}$ on $\mathcal{V}$; moreover, we have $\mathcal{Q} = \{q \in \Delta_c : \operatorname{supp}(q) = \mathcal{K}\} \subsetneq \operatorname{dom} h$ and $\nabla h(q) = 1 + q \log q$ on $\mathcal{Q}$. In the finite-dimensional case, this regularizer forms the basis of the well-known Hedge (or multiplicative/exponential weights) algorithm [5, 6, 39, 54]; for the infinite-dimensional case, see [38, 43] (and below). ◀

With all this in hand, the dual averaging algorithm can be described by means of the abstract recursion

$$y_{t+1} = y_t - \hat{\ell}_t, \quad p_{t+1} = Q(\eta_{t+1} y_{t+1}), \tag{DA}$$

where (*i*) $t = 1, 2, \ldots$ denotes the stage of the process (with the convention $y_0 = \hat{\ell}_0 = 0$); (*ii*) $p_t \in \mathcal{Q}$ is the learner's strategy at stage $t$; (*iii*) $\hat{\ell}_t \in \mathcal{L}^\infty(\mathcal{K})$ is the inexact model revealed at stage $t$; (*iv*) $y_t \in \mathcal{L}^\infty(\mathcal{K})$ is a "score" variable that aggregates loss models up to stage $t$; (*v*) $\eta_t > 0$ is a "learning rate" sequence; and (*vi*) $Q\colon \mathcal{L}^\infty(\mathcal{K}) \to \mathcal{Q}$ is the method's *mirror map*, viz.

$$Q(\psi) = \arg\max_{p \in \mathcal{V}} \{\langle \psi, p \rangle - h(p)\}. \tag{10}$$

For a pseudocode implementation, see Alg. 1 below. In the paper's supplement we also show that the method is *well-posed*, i.e., the $\arg\max$ in (10) is attained at a valid prox-strategy $p_t \in \mathcal{Q}$. We illustrate this with an example:

▶ **Example 6** (Logit choice). Suppose that $h(p) = \int_{\mathcal{K}} p \log p$ is the entropic regularizer of Example 5. Then, the corresponding mirror map is given in closed form by the logit choice model:

$$\Lambda(\varphi) = \frac{\exp(\varphi)}{\int_{\mathcal{K}} \exp(\varphi)} \quad \text{for all } \varphi \in \mathcal{L}^\infty(\mathcal{K}). \tag{11}$$

This derivation builds on a series of well-established arguments that we defer to the supplement. Clearly, $\int_{\mathcal{K}} \Lambda(\varphi) = 1$ and $\Lambda(\varphi) > 0$ as a function on $\mathcal{K}$, so $\Lambda(\varphi)$ is a valid prox-strategy. ◀

| **Algorithm 1:** Dual averaging with imperfect feedback | [Hedge variant: $Q \leftarrow \Lambda$] |
|---|---|

```
Require: mirror map Q: L^∞(K) → Q; learning rate η_t > 0; initialize: y_1 ← 0
1: for t = 1, 2, ... do
2:     set p_t ← Q(η_t y_t)      [p_t ← Λ(η_t y_t) for Hedge]      # update mixed strategy
3:     play x_t ∼ p_t                                              # choose action
4:     observe ℓ̂_t                                                 # model revealed
5:     set y_{t+1} ← y_t − ℓ̂_t                                     # update scores
6: end for
```

## 4 General regret bounds

**4.1. Static regret guarantees.**   We are now in a position to state our first result for (DA):

**Proposition 1.** *For any simple strategy $\chi \in \mathcal{X}$, Alg. 1 enjoys the bound*

$$\text{Reg}_\chi(T) \le \eta_{T+1}^{-1}[h(\chi) - \min h] + \sum_{t=1}^T \langle e_t, \chi - p_t \rangle + \frac{1}{2K} \sum_{t=1}^T \eta_t \|\hat{\ell}_t\|_*^2. \tag{12}$$

Proposition 1 is a "template" bound that we will use to extract static and dynamic regret guarantees in the sequel. Its proof relies on a suitable energy function measuring the match between the learner's aggregate model $y_t$ and the comparator $\chi$. The main difficulty is that these variables live in completely different spaces ($\mathcal{L}^\infty(\mathcal{K})$ vs. $\mathcal{X}$ respectively), so there is no clear distance metric connecting them. However, since bounded functions $\psi \in \mathcal{L}^\infty(\mathcal{K})$ and simple strategies $\chi \in \mathcal{X}$ are naturally paired via duality, they are indirectly connected via the Fenchel–Young inequality $\langle \psi, \chi \rangle \le h(\chi) + h^*(\psi)$, where

$$h^*(\psi) = \max_{p \in \mathcal{V}} \{ \langle \psi, p \rangle - h(p) \} \tag{13}$$

denotes the convex conjugate of $h$. We will thus consider the energy function

$$E_t := \eta_t^{-1}[h(\chi) + h^*(\eta_t y_t) - \langle \eta_t y_t, \chi \rangle]. \tag{14}$$

By construction, $E_t \ge 0$ for all $t$ and $E_t = 0$ if and only if $p_t = Q(\eta_t y_t) = \chi$. More to the point, the defining property of $E_t$ is the following recursive bound (which we prove in the supplement):

**Lemma 2.** *For all $\chi \in \mathcal{X}$, we have:*

$$E_{t+1} \le E_t + \langle \hat{\ell}_t, \chi - p_t \rangle + (\eta_{t+1}^{-1} - \eta_t^{-1})[h(\chi) - \min h] + \frac{\eta_t}{2K} \|\hat{\ell}_t\|_*^2. \tag{15}$$

Proposition 1 is obtained by telescoping (15); subsequently, to obtain a regret bound for Alg. 1, we must relate $\text{Reg}_x(T)$ to $\text{Reg}_\chi(T)$. This can be achieved by invoking Lemma 1 but the resulting expressions are much simpler when $h$ is *decomposable*, i.e., $h(p) = \int_\mathcal{K} \theta(p(x)) \, dx$ for some $C^2$ function $\theta: [0, \infty) \to \mathbb{R}$ with $\theta'' > 0$. In this more explicit setting, we have:

**Theorem 1.** *Fix $x \in \mathcal{K}$, let $\mathcal{C}$ be a convex neighborhood of $x$ in $\mathcal{K}$, and suppose that Alg. 1 is run with a decomposable regularizer $h(p) = \int_\mathcal{K} \theta \circ p$. Then, letting $\phi(z) = z\theta(1/z)$ for $z > 0$, we have:*

$$\mathbb{E}[\text{Reg}_x(T)] \le \frac{\phi(\lambda(\mathcal{C})) - \phi(\lambda(\mathcal{K}))}{\eta_{T+1}} + L \, \text{diam}(\mathcal{C})T + 2 \sum_{t=1}^T B_t + \frac{\alpha^2}{2K} \sum_{t=1}^T \eta_t M_t^2. \tag{16}$$

*In particular, if Alg. 1 is run with learning rate $\eta_t \propto 1/t^\rho$, $\rho \in (0, 1)$, and inexact models such that $B_t = \mathcal{O}(1/t^\beta)$ and $M_t^2 = \mathcal{O}(t^{2\mu})$ for some $\beta, \mu \ge 0$, we have:*

$$\mathbb{E}[\text{Reg}(T)] = \mathcal{O}(\phi(T^{-n\kappa})T^\rho + T^{1-\kappa} + T^{1-\beta} + T^{1+2\mu-\rho}) \quad \text{for all } \kappa \ge 0. \tag{17}$$

**Corollary 1.** *If the learner's feedback is unbiased and bounded in mean square (i.e., $B_t = 0$ and $\sup_t M_t < \infty$), running Alg. 1 with learning rate $\eta_t \propto 1/t^\rho$ guarantees*

$$\mathbb{E}[\text{Reg}(T)] = \mathcal{O}(\phi(T^{-n\rho})T^\rho + T^{1-\rho}). \tag{18}$$

*In particular, for the regularizers of Examples 4 and 5, we have:*

1. *For $\theta(z) = (1/2)z^2$, Alg. 1 with $\eta_t \propto t^{-1/(n+2)}$ guarantees $\mathbb{E}[\text{Reg}(T)] = \mathcal{O}(T^{\frac{n+1}{n+2}})$.*

2. *For $\theta(z) = z \log z$, Alg. 1 with $\eta_t \propto t^{-1/2}$ guarantees $\mathbb{E}[\text{Reg}(T)] = \mathcal{O}(T^{1/2})$.*

*Remark* 2. Here and in the sequel, logarithmic factors are ignored in the Landau $\mathcal{O}(\cdot)$ notation. We should also stress that the role of $\mathcal{C}$ in Theorem 1 only has to do with the analysis of the algorithm, not with the derived bounds (which are obtained by picking a suitable $\mathcal{C}$).

First, in online *convex* optimization, dual averaging with stochastic gradient feedback achieves $\mathcal{O}(\sqrt{T})$ regret *irrespective* of the choice of regularizer, and this bound is tight [1, 16, 45]. By contrast, in the non-convex setting, the choice of regularizer has a visible impact on the regret because it affects the exponent of $T$: in particular, $L^2$ regularization carries a much worse dependence on $T$ relative to the Hedge variant of Alg. 1. This is due to the term $\mathcal{O}(\phi(T^{-n\kappa})T^\rho)$ that appears in (17) and is in turn linked to the choice of the "enclosure set" $\mathcal{C}$ having $\lambda(\mathcal{C}) \propto T^{-n\kappa}$ for some $\kappa \geq 0$.

The negentropy regularizer (and any other regularizer with quasi-linear growth at infinity, see the supplement for additional examples) only incurs a logarithmic dependence on $\lambda(\mathcal{C})$. Instead, the quadratic growth of the $L^2$ regularizer induces an $\mathcal{O}(1/\lambda(\mathcal{C}))$ term in the algorithm's regret, which is ultimately responsible for the catastrophic dependence on the dimension of $\mathcal{K}$. Seeing as the bounds achieved by the Hedge variant of Alg. 1 are optimal in this regard, we will concentrate on this specific instance in the sequel.

**4.2. Dynamic regret guarantees.** We now turn to the dynamic regret minimization guarantees of Alg. 1. In this regard, we note first that, in complete generality, dynamic regret minimization is *not possible* because an informed adversary can always impose a uniformly positive loss at each stage [45]. Because of this, dynamic regret guarantees are often stated in terms of the *variation* of the loss functions encountered, namely

$$V_T \coloneqq \sum_{t=1}^{T} \|\ell_{t+1} - \ell_t\|_\infty = \sum_{t=1}^{T} \max_{x \in \mathcal{K}} |\ell_{t+1}(x) - \ell_t(x)|, \tag{19}$$

with the convention $\ell_{t+1} = \ell_t$ for $t = T$.[3] We then have:

**Theorem 2.** *Suppose that the Hedge variant of Alg. 1 is run with learning rate $\eta_t \propto 1/t^\rho$ and inexact models with $B_t = \mathcal{O}(1/t^\beta)$ and $M_t^2 = \mathcal{O}(t^{2\mu})$ for some $\beta, \mu \geq 0$. Then:*

$$\mathbb{E}[\mathrm{DynReg}(T)] = \mathcal{O}(T^{1+2\mu-\rho} + T^{1-\beta} + T^{2\rho-2\mu}V_T). \tag{20}$$

*In particular, if $V_T = \mathcal{O}(T^\nu)$ for some $\nu < 1$ and the learner's feedback is unbiased and bounded in mean square (i.e., $B_t = 0$ and $\sup_t M_t < \infty$), the choice $\rho = (1-\nu)/3$ guarantees*

$$\mathbb{E}[\mathrm{DynReg}(T)] = \mathcal{O}(T^{\frac{2+\nu}{3}}). \tag{21}$$

To the best of our knowledge, Theorem 2 provides the first dynamic regret guarantee for online non-convex problems. The main idea behind its proof is to examine the evolution of play over a series of windows of length $\Delta = \mathcal{O}(T^\gamma)$ for some $\gamma > 0$. In so doing, Theorem 1 can be used to obtain a bound for the learner's regret relative to the best action $x \in \mathcal{K}$ *within each window*. Obviously, if the length of the window is chosen sufficiently small, aggregating the learner's regret *per window* will be a reasonable approximation of the learner's dynamic regret. At the same time, if the window is taken too small, the number of such windows required to cover $T$ will be $\Theta(T)$, so this approximation becomes meaningless. As a result, to obtain a meaningful regret bound, this window-by-window examination of the algorithm must be carefully aligned with the variation $V_T$ of the loss functions encountered by the learner. Albeit intuitive, the details required to make this argument precise are fairly subtle, so we relegate the proof of Theorem 2 to the paper's supplement.

We should also observe here that the $\mathcal{O}(T^{\frac{2+\nu}{3}})$ bound of Theorem 2 is, in general, unimprovable, even if the losses are *linear*. Specifically, Besbes et al. [11] showed that, if the learner is facing a stream of linear losses with stochastic gradient feedback (i.e., an inexact linear model), an informed adversary can still impose $\mathrm{DynReg}(T) = \Omega(T^{2/3}V_T^{1/3})$. Besbes et al. [11] further proposed a scheme to achieve this bound by means of a periodic restart meta-principle that partitions the horizon of play into batches of size $(T/V_T)^{2/3}$ and then runs an algorithm achieving $(T/V_T)^{1/3}$ regret per batch. Theorem 2 differs from the results of Besbes et al. [11] in two key aspects: (*a*) Alg. 1 does not require a periodic restart schedule (so the learner does not forget the information accrued up to a given stage); and (*b*) more importantly, it applies to *general* online optimization problems, without a convex structure or any other structural assumptions (though with a different feedback structure).

**Algorithm 2:** Bandit dual averaging                                         [Hedge variant: $Q \leftarrow \Lambda$]

---
**Require:** mirror map $Q: \mathcal{L}^\infty(\mathcal{K}) \to \mathcal{Q}$; parameters $\eta_t, \delta_t, \varepsilon_t > 0$; **initialize:** $y_1 \leftarrow 0$
1: **for** $t = 1, 2, \ldots$ **do**
2:     set $p_t \leftarrow (1 - \varepsilon_t)Q(\eta_t y_t) + \varepsilon_t/\lambda(\mathcal{K})$     [$Q \leftarrow \Lambda$ for Hedge]          # mixed strategy
3:     play $x_t \sim p_t$                                                                                    # choose action
4:     set $\hat{\ell}_t = K^{\delta_t}(x_t, \cdot) \cdot \ell_t(x_t)/p_t(x_t)$                                              # payoff model
5:     set $y_{t+1} \leftarrow y_t - \hat{\ell}_t$                                                                   # update scores
6: **end for**

---

## 5   Applications to online non-convex learning with bandit feedback

As an application of the inexact model framework of the previous sections, we proceed to consider the case where the learner only observes their realized reward $\ell_t(x_t)$ and has *no other information*. In this "bandit setting", an inexact model is not available and must instead be constructed on the fly.

When $\mathcal{K}$ is a finite set, $\ell_t$ is a $|\mathcal{K}|$-dimensional vector, and an unbiased estimator for $\ell_t$ can be constructed by setting $\hat{\ell}_t(x) = [\mathbb{1}\{x = x_t\}/\mathbb{P}(x = x_t)]\ell_t(x_t)$ for all $x \in \mathcal{K}$. This "importance weighted" estimator is the basis for the EXP3 variant of the Hedge algorithm which is known to achieve $\mathcal{O}(T^{1/2})$ regret [8]. However, in the case of *continuous* action spaces, there is a key obstacle: if the indicator $\mathbb{1}\{x = x_t\}$ is replaced by a Dirac point mass $\delta_{x_t}(x)$, the resulting loss model $\hat{\ell}_t \propto \delta_{x_t}$ would no longer be a function but a generalized (singular) distribution, so the dual averaging framework of Alg. 1 no longer applies.

To counter this, we will take a "smoothing" approach in the spirit of [19] and consider the estimator

$$\hat{\ell}_t(x) = K_t(x_t, x) \cdot \ell_t(x_t)/p_t(x_t) \tag{22}$$

where $K_t: \mathcal{K} \times \mathcal{K} \to \mathbb{R}$ is a (time-varying) *smoothing kernel*, i.e., $\int_\mathcal{K} K_t(x, x') \, dx' = 1$ for all $x \in \mathcal{K}$. For concreteness (and sampling efficiency), we will assume that losses now take values in $[0, 1]$, and we will focus on simple kernels that are supported on a neighborhood $\mathcal{U}_\delta(x) = \mathbb{B}_\delta(x) \cap \mathcal{K}$ of $x$ in $\mathcal{K}$ and are constant therein, i.e., $K^\delta(x, x') = [\lambda(\mathcal{U}_\delta(x))]^{-1}\mathbb{1}\{\|x' - x\| \le \delta\}$.

The "smoothing radius" $\delta$ in the definition of $K^\delta$ will play a key role in the choice of loss model being fed to Alg. 1. If $\delta$ is taken too small, $K^\delta$ will approach a point mass, so it will have low estimation error but very high variance; at the other end of the spectrum, if $\delta$ is taken too large, the variance of the induced estimator will be low, but so will its accuracy. In view of this, we will consider a flexible smoothing schedule of the form $\delta_t = 1/t^\mu$ which gradually sharpens the estimator over time as more information comes in. Then, to further protect the algorithm from getting stuck in local minima, we will also incorporate in $p_t$ an explicit exploration term of the form $\varepsilon_t/\lambda(\mathcal{K})$.

Putting all this together, we obtain the *bandit dual averaging* (BDA) algorithm presented in pseudocode form as Alg. 2 above. By employing a slight variation of the analysis presented in Section 4 (basically amounting to a tighter bound in Lemma 2), we obtain the following guarantees for Alg. 2:

**Proposition 2.** *Suppose that the Hedge variant of Alg. 2 is run with learning rate $\eta_t \propto 1/t^\rho$ and smoothing/exploration schedules $\delta_t \propto 1/t^\mu$, $\varepsilon_t \propto 1/t^\beta$ respectively. Then, the learner enjoys the bound*

$$\mathbb{E}[\mathrm{Reg}(T)] = \mathcal{O}(T^\rho + T^{1-\mu} + T^{1-\beta} + T^{1+n\mu+\beta-\rho}). \tag{23}$$

*In particular, if the algorithm is run with $\rho = (n + 2)/(n + 3)$ and $\mu = \beta = 1/(n + 3)$, we obtain the bound $\mathbb{E}[\mathrm{Reg}(T)] = \mathcal{O}(T^{\frac{n+2}{n+3}})$.*

**Proposition 3.** *Suppose that the Hedge variant of Alg. 2 is run with parameters as in Proposition 2 against a stream of loss functions with variation $V_T = \mathcal{O}(T^\nu)$. Then, the learner enjoys*

$$\mathbb{E}[\mathrm{DynReg}(T)] = \mathcal{O}(T^{1+n\mu+\beta-\rho} + T^{1-\beta} + T^{1-\mu} + T^{\nu+2\rho-n\mu-\beta}). \tag{24}$$

*In particular, if the algorithm is run with $\rho = (1 - \nu)(n + 2)/(n + 4)$ and $\mu = \beta = (1 - \nu)/(n + 4)$, we obtain the optimized bound $\mathbb{E}[\mathrm{DynReg}(T)] = \mathcal{O}(T^{\frac{n+3+\nu}{n+4}})$.*

To the best of our knowledge, Proposition 3 is the first result of its kind for dynamic regret minimization in online non-convex problems with bandit feedback. We conjecture that the bounds of Propositions 2 and 3 can be tightened further to $\mathcal{O}(T^{\frac{n+1}{n+2}})$ and $\mathcal{O}(T^{\frac{n+2+\nu}{n+3}})$ by dropping the explicit exploration term; we defer this finetuning to future work.

## Acknowledgments

P. Mertikopoulos is grateful for financial support by the French National Research Agency (ANR) in the framework of the "Investissements d'avenir" program (ANR-15-IDEX-02), the LabEx PER-SYVAL (ANR-11-LABX-0025-01), and MIAI@Grenoble Alpes (ANR-19-P3IA-0003). This research was also supported by the COST Action CA16228 "European Network for Game Theory" (GAMENET).

## Broader Impact

This is a theoretical work which does not present any foreseeable societal consequence.

## Footnotes

[1] Since the dual space of $\mathcal{M}(\mathcal{K})$ contains $\mathcal{L}^\infty(\mathcal{K})$, we will also view $\mathcal{L}^\infty(\mathcal{K})$ as an embedded subset of $\mathcal{V}^*$.

[2] In this case, $\alpha = \sqrt{\lambda(\mathcal{K})}$: this is because $\|p\|_{\mathrm{TV}}^2 = \left[\int_{\mathcal{K}} p\right]^2 \leq \int_{\mathcal{K}} p^2 \cdot \int_{\mathcal{K}} 1 = \lambda(\mathcal{K})\|p\|_2^2$ if $p \in \Delta_c$.

[3]This notion is due to Besbes et al. [11]. Other notions of variation have also been considered [11, 21, 23], as well as other measures of regret, cf. [30, 32]; for a survey, see [20].

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
