[Supplementary Material]

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

# A Examples

In this appendix, we provide some more decomposable regularizers that are commonly used in the literature:

▶ **Example 7** (Log-barrier regularization). Let $\mathcal{V} = \mathcal{M}(\mathcal{K})$ as above and consider the so-called *Burg entropy* $h(p) = -\int_{\mathcal{K}} \log p$ [4]. In this case, $\mathcal{Q} = \mathrm{dom}\,\partial h = \{q \in \Delta_c : \mathrm{supp}(q) = \mathcal{K}\} = \mathrm{dom}\,h$ and $\nabla h(q) = -1/q$ on $\mathcal{Q}$. In the finite-dimensional case, this regularizer plays a fundamental role in the affine scaling method of Karmarkar [35], see e.g., Tseng [52], Vanderbei et al. [53] and references therein. The corresponding mirror map is obtained as follows: let $L(p;\lambda) = \int_{\mathcal{K}} \psi p + \int_{\mathcal{K}} \log p - \lambda \int_{\mathcal{K}} p$ denote the Lagrangian of the problem (10), so $q = Q(\psi)$ satisfies the first-order optimality condition

$$\psi + 1/q - \lambda = 0. \tag{A.1}$$

Solving for $q$ and integrating, we get $\int_{\mathcal{K}}(\lambda - \psi)^{-1} = \int_{\mathcal{K}} q = 1$. The function $\phi(\lambda) = \int_{\mathcal{K}}(\lambda - \psi)^{-1}$ is decreasing in $\lambda$ and continuous whenever finite; moreover, since $\psi \in \mathcal{L}^\infty(\mathcal{K})$, it follows that $\phi$ is always finite (and hence continuous) for large enough $\lambda$, and $\lim_{\lambda \to \infty} \phi(\lambda) = 0$. Since $\sup_\lambda \phi(\lambda) = \infty$, there exists some maximal $\lambda^*$ such that (A.1) holds (in practice, this can be located by a simple line search initialized at some $\lambda > \|\psi\|_\infty$). We thus get $Q(\psi) = (\lambda^* - \psi)^{-1}$. ◀

▶ **Example 8** (Tsallis entropy). A generalization of the Shannon-Gibbs entropy for nonextensive variables is the *Tsallis entropy* [51] defined here as $h(p) = \int_{\mathcal{K}} \theta(p)$ where $\theta(z) = [\gamma(1-\gamma)]^{-1}(z-z^\gamma)$ for $\gamma \in (0,1]$, with the continuity convention $(z - z^\gamma)/(1-\gamma) = z \log z$ for $\gamma = 1$ (corresponding to the Shannon-Gibbs case). Working as in Example 7, we have $\mathcal{Q} = \mathrm{dom}\,\partial h = \{q \in \Delta_c : \mathrm{supp}(q) = \mathcal{K}\} \subsetneq \mathrm{dom}\,h$, and the corresponding mirror map $q = Q(\psi)$ is obtained via the first-order stationarity equation

$$\psi - \frac{1 - \gamma q^{\gamma-1}}{\gamma(1-\gamma)} - \lambda = 0. \tag{A.2}$$

Then, solving for $q$ yields $Q(\psi) = (1-\gamma)^{1/(\gamma-1)} \int_{\mathcal{K}}(\mu - \psi)^{1/(\gamma-1)}$ with $\mu > \|\psi\|_\infty$ chosen so that $\int_{\mathcal{K}} Q(\psi) = 1$. ◀

# B Basic properties of regularizers and mirror maps

The goal of this appendix is to prove some basic results on regularizer functions and mirror maps that will be used liberally in the sequel. Versions of the results presented here already exist in the literature, but our infinite-dimensional setting introduces some subtleties that require further care. For this reason, we state and prove all required results for completeness.

We begin by recalling some definitions from the main part of the paper. First, we write $\mathcal{M} \equiv \mathcal{M}(\mathcal{K})$ for the space of all finite signed Radon measures on $\mathcal{K}$ equipped with the total variation norm $\|\mu\|_{\mathrm{TV}} = \mu^+(\mathcal{K}) + \mu^-(\mathcal{K})$, where $\mu^+$ (resp. $\mu^-$) denotes the positive (resp. negative) part of $\mu$ coming from the Hahn-Banach decomposition of signed measures on $\mathcal{K}$. As we discussed in Section 3, we also assume given a model Banach space $\mathcal{V}$ containing the set of simple strategies $\mathcal{X}$ as an embedded subset and such that $\|\cdot\|_{\mathrm{TV}} \leq \alpha\|\cdot\|$ for some $\alpha > 0$.

With all this in hand, we begin by discussing the well-posedness of Alg. 1. To that end, we have the following basic result:

**Lemma B.1.** *Let $h$ be a regularizer on $\mathcal{X}$. Then:*

1. *$Q(\psi) \in \mathcal{Q}$ for all $\psi \in \mathcal{V}^*$; in particular:*

$$q = Q(\psi) \iff \psi \in \partial h(q). \tag{B.1}$$

2. *If $q = Q(\psi)$ and $p \in \mathrm{dom}\,h$, we have*

$$\langle \nabla h(q), q - p \rangle \leq \langle \psi, q - p \rangle. \tag{B.2}$$

3. *The convex conjugate $h^*(\psi) = \max_{p \in \mathcal{V}}\{\langle \psi, p \rangle - h(p)\}$ is Fréchet differentiable and satisfies*

$$\mathrm{D}_v\, h^*(\psi) = \langle v, Q(\psi) \rangle \quad \text{for all } \psi, v \in \mathcal{V}^*. \tag{B.3}$$

**Corollary 2.** *Alg. 1 is well-posed, i.e., $p_t \in \mathcal{Q}$ for all $t = 1, 2, \dots$ if $\hat{\ell}_t \in \mathcal{L}^\infty(\mathcal{K})$.*

*Proof.* We proceed item by item:

1. First, since $h$ is strongly convex and lower semi-continuous, the maximum in (10) is attained. Hence, by Fermat's rule for subdifferentials, $q$ solves (10) if and only if $\psi - \partial h(q) \ni 0$. We thus get the string of equivalences:

$$q = Q(\psi) \iff \psi - \partial h(q) \ni 0 \iff \psi \in \partial h(q). \tag{B.4}$$

   In particular, this implies that $\psi \in \operatorname{dom} \partial h(q) \neq \varnothing$, i.e., $q \in \operatorname{dom} \partial h =: \mathcal{Q}$, as claimed.

2. To establish (B.2), it suffices to show that it holds for all $q \in \mathcal{Q}$ (by continuity). To do so, let

$$\phi(t) = h(q + t(p - q)) - [h(q) + \langle \psi, q + t(p - q) \rangle]. \tag{B.5}$$

   Since $h$ is strongly convex relative $\psi \in \partial h(q)$ by (B.1), it follows that $\phi(t) \geq 0$ with equality if and only if $t = 0$. Moreover, note that $\psi(t) = \langle \nabla h(q + t(p - q)) - \psi, p - q \rangle$ is a continuous selection of subgradients of $\phi$. Given that $\phi$ and $\psi$ are both continuous on $[0, 1]$, it follows that $\phi$ is continuously differentiable and $\phi' = \psi$ on $[0, 1]$. Thus, with $\phi$ convex and $\phi(t) \geq 0 = \phi(0)$ for all $t \in [0, 1]$, we conclude that $\phi'(0) = \langle \nabla h(q) - \psi, p - q \rangle \geq 0$, from which our claim follows.

Finally, the Fréchet differentiability of $h^*$ is a straightforward application of the envelope theorem, which is sometimes referred to in the literature as Danskin's theorem, cf. Berge [10, Chap. 4] □

As we mentioned in the main text, much of our analysis revolves around the energy function (14) defined by means of the Fenchel-Young inequality. To formalize this, it will be convenient to introduce a more general pairing between $p \in \mathcal{V}$ and $\psi \in \mathcal{V}^*$, known as the *Fenchel coupling*. Following [40], this is defined as

$$F(p, \psi) = h(p) + h^*(\psi) - \langle \psi, p \rangle \quad \text{for all } p \in \operatorname{dom} h, \psi \in \mathcal{V}^*. \tag{B.6}$$

The following series of lemmas gathers some basic properties of the Fenchel coupling. The first is a lower bound for the Fenchel coupling in terms of the ambient norm in $\mathcal{V}$:

**Lemma B.2.** *Let $h$ be a regularizer on $\mathcal{X}$ with strong convexity modulus $K$. Then, for all $p \in \operatorname{dom} h$ and all $\psi \in \mathcal{V}^*$, we have*

$$F(p, \psi) \geq \frac{K}{2} \|Q(\psi) - p\|^2. \tag{B.7}$$

*Proof.* By the definition of $F$ and the inequality (B.2), we have:

$$
\begin{aligned}
F(p, \psi) &= h(p) + h^*(\psi) - \langle \psi, p \rangle = h(p) + \langle \psi, Q(\psi) \rangle - h(Q(\psi)) - \langle \psi, p \rangle \\
&\geq h(p) - h(Q(\psi)) - \langle \nabla h(\psi), Q(\psi) - p \rangle \\
&\geq \frac{K}{2} \|Q(\psi) - p\|^2
\end{aligned}
\tag{B.8}
$$

where we used (B.2) in the second line, and the strong convexity of $h$ in the last. □

Our next result is the primal-dual analogue of the so-called "three-point identity" for the Bregman divergence [22]:

**Proposition B.1.** *Let $h$ be a regularizer on $\mathcal{X}$, fix some $p \in \mathcal{V}$, $\psi, \psi^+ \in \mathcal{V}^*$, and let $q = Q(\psi)$. Then:*

$$F(p, \psi^+) = F(p, \psi) + F(q, \psi^+) + \langle \psi^+ - \psi, q - p \rangle. \tag{B.9}$$

*Proof.* By definition:

$$
\begin{aligned}
F(p, \psi^+) &= h(p) + h^*(\psi^+) - \langle \psi^+, p \rangle \\
F(p, \psi) &= h(p) + h^*(\psi) - \langle \psi, p \rangle.
\end{aligned}
\tag{B.10}
$$

Thus, by subtracting the above, we get:

$$
\begin{aligned}
F(p, \psi^+) - F(p, \psi) &= h(p) + h^*(\psi^+) - \langle \psi^+, p \rangle - h(p) - h^*(\psi) + \langle \psi, p \rangle \\
&= h^*(\psi^+) - h^*(\psi) - \langle \psi^+ - \psi, p \rangle \\
&= h^*(\psi^+) - \langle \psi, Q(\psi) \rangle + h(Q(\psi)) - \langle \psi^+ - \psi, p \rangle \\
&= h^*(\psi^+) - \langle \psi, q \rangle + h(q) - \langle \psi^+ - \psi, p \rangle \\
&= h^*(\psi^+) + \langle \psi^+ - \psi, q \rangle - \langle \psi^+, q \rangle + h(q) - \langle \psi^+ - \psi, p \rangle \\
&= F(q, \psi^+) + \langle \psi^+ - \psi, q - p \rangle
\end{aligned}
\tag{B.11}
$$

and our proof is complete. $\qquad\square$

We are now in a position to state and prove a key inequality for the Fenchel coupling:

**Proposition B.2.** *Let $h$ be a regularizer on $\mathcal{X}$ with convexity modulus $K$, fix some $p \in \mathrm{dom}\, h$, and let $q = Q(\psi)$ for some $\psi \in \mathcal{V}^*$. Then, for all $v \in \mathcal{V}^*$, we have:*

$$
F(p, \psi + v) \leq F(p, \psi) + \langle v, q - p \rangle + \frac{1}{2K} \|v\|_*^2
\tag{B.12}
$$

*Proof.* Let $q = Q(\psi)$, $\psi^+ = \psi + v$, and $q^+ = Q(\psi^+)$. Then, by the three-point identity (B.9), we have

$$
F(p, \psi) = F(p, \psi^+) + F(q^+, \psi) + \langle \psi - \psi^+, q^+ - p \rangle.
\tag{B.13}
$$

Hence, after rearranging:

$$
\begin{aligned}
F(p, \psi^+) &= F(p, \psi) - F(q^+, \psi) + \langle v, q^+ - p \rangle \\
&= F(p, \psi) - F(q^+, \psi) + \langle v, q - p \rangle + \langle v, q^+ - q \rangle.
\end{aligned}
\tag{B.14}
$$

By Young's inequality, we also have

$$
\langle v, q^+ - q \rangle \leq \frac{K}{2} \|q^+ - q\|^2 + \frac{1}{2K} \|v\|_*^2.
\tag{B.15}
$$

Thus, substituting in (B.14), we get

$$
F(p, \psi^+) \leq F(p, \psi) + \langle v, q - p \rangle + \frac{1}{2K} \|v\|_*^2 - F(q^+, \psi) + \frac{K}{2} \|q^+ - q\|^2.
\tag{B.16}
$$

Our claim then follows by noting that $F(q^+, \psi) \geq \frac{K}{2} \|q^+ - q\|^2$ (cf. Lemma B.2 above). $\qquad\square$

## C  Regret derivations

**Notation: from losses to payoffs.**   In this appendix, we prove the general regret guarantees for Alg. 1. For notational convenience, we will switch in what follows from "losses" to "payoffs", i.e., we will assume that the learner is encountering a sequence of payoff functions $u_t = -\ell_t$ and gets as feedback the model $\hat{u}_t = -\hat{\ell}_t$.

**C.1. Basic bounds and preliminaries.**   We begin by providing some template regret bounds that we will use as a toolkit in the sequel. As a warm-up, we prove the basic comparison lemma between simple and pure strategies:

**Lemma 1.** *Let $\mathcal{U}$ be a convex neighborhood of $x$ in $\mathcal{K}$ and let $\chi \in \mathcal{X}$ be a simple strategy supported on $\mathcal{U}$. Then, $\mathrm{Reg}_x(T) \leq \mathrm{Reg}_\chi(T) + L\,\mathrm{diam}(\mathcal{U})T$.*

*Proof.* By Assumption 1, we have $u_t(x) \leq u_t(x') + L\|x - x'\| \leq u_t(x') + L\,\mathrm{diam}(\mathcal{U})$ for all $x' \in \mathcal{U}$. Hence, taking expectations on both sides relative to $\chi$, we get $u_t(x) \leq \langle u_t, \chi \rangle + L\,\mathrm{diam}(\mathcal{U})$. Our claim then follows by summing over $t = 1, 2, \ldots, T$ and invoking the definition of the regret. $\qquad\square$

We now turn to the derivation of our main regret guarantees as outlined in Section 4. Much of the analysis to follow will revolve around the energy function (14) which, for convenience, we restate below in terms of the Fenchel coupling (B.6):

$$E_t := \frac{1}{\eta_t}[h(\chi) + h^*(\eta_t y_t) - \langle \eta_t y_t, \chi \rangle] = \frac{1}{\eta_t} F(\chi, \eta_t y_t). \tag{14}$$

In words, $E_t$ essentially measures the primal-dual "distance" between the benchmark strategy $\chi$ and the aggregate model $y_t$, taking into account the inflation of the latter by $\eta_t$ in (DA). Our overall proof strategy will then be to relate the regret incurred by the optimizer to the evolution of $E_t$ over time. To that end, an application of Abel's summation formula gives:

$$E_{t+1} - E_t = \frac{1}{\eta_{t+1}} F(\chi, \eta_{t+1} y_{t+1}) - \frac{1}{\eta_t} F(\chi, \eta_t y_t)$$

$$= \frac{1}{\eta_{t+1}} F(\chi, \eta_{t+1} y_{t+1}) - \frac{1}{\eta_t} F(\chi, \eta_t y_{t+1}) \tag{C.1a}$$

$$+ \frac{1}{\eta_t} F(\chi, \eta_t y_{t+1}) - \frac{1}{\eta_t} F(\chi, \eta_t y_t). \tag{C.1b}$$

We now proceed to unpack the two terms (C.1a) and (C.1b) separately, beginning with the latter.

To do so, substituting $p \leftarrow \chi$, $\psi \leftarrow \eta_t y_t$ and $\psi^+ \leftarrow \eta_t y_{t+1}$ in Proposition B.1 yields

$$(\text{C.1b}) = \frac{1}{\eta_t}[F(\chi, \eta_t y_t + \eta_t \hat{u}_t) - F(\chi, \eta_t y_t)]$$

$$= \frac{1}{\eta_t}[F(p_t, \eta_t y_{t+1}) + \langle \eta_t \hat{u}_t, p_t - \chi \rangle]$$

$$= \frac{F(p_t, \eta_t y_{t+1})}{\eta_t} + \langle \hat{u}_t, p_t - \chi \rangle \tag{C.2}$$

where we used the definition $p_t = Q(\eta_t y_t)$ of $p_t$. We thus obtain the interim expression

$$E_{t+1} = E_t + (\text{C.1a}) + \langle \hat{u}_t, p_t - \chi \rangle + \frac{F(p_t, \eta_t y_{t+1})}{\eta_t} \tag{C.3}$$

Moving forward, for the term (C.1a), the definition of the Fenchel coupling (B.6) readily yields:

$$(\text{C.1a}) = \left[\frac{1}{\eta_{t+1}} - \frac{1}{\eta_t}\right] h(\chi) + \frac{1}{\eta_{t+1}} h^*(\eta_{t+1} y_{t+1}) - \frac{1}{\eta_t} h^*(\eta_t y_{t+1}). \tag{C.4}$$

Consider now the function $\varphi(\eta) = \eta^{-1}[h^*(\eta \psi) + \min h]$ for arbitrary $\psi \in \mathcal{L}^\infty(\mathcal{K})$. By Lemma B.1, $h^*$ is Fréchet differentiable with $D_v h^*(\cdot) = \langle v, Q(\cdot) \rangle$ for all $v \in \mathcal{V}^*$, so a simple differentiation yields

$$\varphi'(\eta) = \frac{1}{\eta}\langle \psi, Q(\eta \psi) \rangle - \frac{1}{\eta^2}[h^*(\eta \psi) + \min h]$$

$$= \frac{1}{\eta^2}[\langle \eta \psi, Q(\eta \psi) \rangle - h^*(\eta \psi) - \min h]$$

$$= \frac{1}{\eta^2}[h(Q(\eta \psi)) - \min h] \geq 0, \tag{C.5}$$

where we used the Fenchel-Young inequality as an equality in the second-to-last line. Since $\eta_{t+1} \leq \eta_t$, the above shows that $\varphi(\eta_t) \geq \varphi(\eta_{t+1})$. Hence, substituting $\psi \leftarrow y_{t+1}$, we ultimately obtain

$$\frac{1}{\eta_{t+1}} h^*(\eta_{t+1} y_{t+1}) - \frac{1}{\eta_t} h^*(\eta_t y_{t+1}) \leq \left[\frac{1}{\eta_t} - \frac{1}{\eta_{t+1}}\right] \min h. \tag{C.6}$$

Therefore, combining (C.3) and (C.6), we have proved the following template bound:

**Lemma C.1.** *For all $\chi \in \mathcal{X}$, the policy* (DA) *enjoys the bound*

$$E_{t+1} \leq E_t + \langle \hat{u}_t, p_t - \chi \rangle + \left(\frac{1}{\eta_{t+1}} - \frac{1}{\eta_t}\right)[h(\chi) - \min h] + \frac{1}{\eta_t} F(p_t, \eta_t y_{t+1}). \tag{C.7}$$

We are now in a position to prove our basic energy inequality (restated below for convenience):

**Lemma 2.** *For all $\chi \in \mathcal{X}$, we have:*

$$E_{t+1} \leq E_t + \langle \hat{u}_t, \chi - p_t \rangle + \left(\eta_{t+1}^{-1} - \eta_t^{-1}\right)[h(\chi) - \min h] + \frac{\eta_t}{2K}\|\hat{u}_t\|_*^2. \tag{15}$$

*Proof.* Going back to Proposition B.2 and setting $p \leftarrow p_t$, $\psi \leftarrow \eta_t y_t$ and $v \leftarrow \eta_t \hat{u}_t$, we get

$$F(p_t, \eta_t y_{t+1}) \leq F(p_t, \eta_t y_t) + \langle \eta_t \hat{u}_t, p_t - p_t \rangle + \frac{\eta_t^2}{2K}\|\hat{u}_t\|_*^2 = \frac{\eta_t^2}{2K}\|\hat{u}_t\|_*^2 \tag{C.8}$$

where we used the fact that $p_t = Q(\eta_t y_t)$. Our claim then follows by dividing both sides by $\eta_t$ and substituting in Lemma C.1. $\square$

We will come back to these results as needed.

**C.2. Static regret guarantees.** We are now ready to prove our static regret results for Alg. 1. We begin with the precursor to our main result in that respect:

**Proposition 1.** *For any simple strategy $\chi \in \mathcal{X}$, Alg. 1 enjoys the bound*

$$\mathrm{Reg}_\chi(T) \leq \eta_{T+1}^{-1}[h(\chi) - \min h] + \sum_{t=1}^T \langle e_t, \chi - p_t \rangle + \frac{1}{2K}\sum_{t=1}^T \eta_t \|\hat{u}_t\|_*^2. \tag{12}$$

*Proof.* Recalling the decomposition $\hat{u}_t = u_t + e_t$ for the learner's inexact models, a simple rearrangement of Lemma 2 gives

$$\langle u_t, \chi - p_t \rangle \leq E_t - E_{t+1} + \langle e_t, p_t - \chi \rangle + \left(\eta_{t+1}^{-1} - \eta_t^{-1}\right)[h(\chi) - \min h] + \frac{\eta_t}{2K}\|\hat{u}_t\|_*^2. \tag{C.9}$$

Thus, telescoping over $t = 1, 2, \ldots, T$, we get

$$\mathrm{Reg}_\chi(T) \leq E_1 - E_{T+1} + \left(\frac{1}{\eta_{T+1}} - \frac{1}{\eta_1}\right)[h(\chi) - \min h] + \sum_{t=1}^T \langle e_t, p_t - \chi \rangle + \frac{1}{2K}\sum_{t=1}^T \eta_t \|\hat{u}_t\|_*^2$$

$$\leq \frac{h(\chi) - \min h}{\eta_{T+1}} + \sum_{t=1}^T \langle e_t, p_t - \chi \rangle + \frac{1}{2K}\sum_{t=1}^T \eta_t \|\hat{u}_t\|_*^2, \tag{C.10}$$

where we used the fact that $E_t \geq 0$ for all $t$ and $E_1 = \eta_1^{-1}[h(\chi) + h^*(0)] = \eta_1^{-1}[h(\chi) - \min h]$. $\square$

As a simple application of Lemma 2, we get the following bound for simple comparators:

**Corollary 3.** *For all $\chi \in \mathcal{X}$, Alg. 1 guarantees*

$$\mathbb{E}[\mathrm{Reg}_\chi(T)] \leq \frac{h(\chi) - \min h}{\eta_{T+1}} + 2\sum_{t=1}^T B_t + \frac{1}{2K}\sum_{t=1}^T \eta_t \,\mathbb{E}[\|\hat{u}_t\|_*^2], \tag{C.11}$$

*Proof.* Simply take expectations over (12) and use the fact that

$$\mathbb{E}[\langle e_t, p_t - \chi \rangle] = \mathbb{E}[\langle \mathbb{E}[e_t \mid \mathcal{F}_t], p_t - \chi \rangle] = \mathbb{E}[\langle b_t, p_t - \chi \rangle\,|] \leq \mathbb{E}[\|b_t\|_\infty \|p_t - \chi\|_1] \leq 2B_t. \quad \square$$

We are finally in a position to prove the main static regret guarantee of Alg. 1:

**Theorem 1.** *Fix $x \in \mathcal{K}$, let $\mathcal{C}$ be a convex neighborhood of $x$ in $\mathcal{K}$, and suppose that Alg. 1 is run with a decomposable regularizer $h(p) = \int_\mathcal{K} \theta \circ p$. Then, letting $\phi(z) = z\theta(1/z)$ for $z > 0$, we have:*

$$\mathbb{E}[\mathrm{Reg}_x(T)] \leq \frac{\phi(\lambda(\mathcal{C})) - \phi(\lambda(\mathcal{K}))}{\eta_{T+1}} + L\,\mathrm{diam}(\mathcal{C})T + 2\sum_{t=1}^T B_t + \frac{\alpha^2}{2K}\sum_{t=1}^T \eta_t M_t^2. \tag{16}$$

*In particular, if Alg. 1 is run with learning rate $\eta_t \propto 1/t^\rho$, $\rho \in (0,1)$, and inexact models such that $B_t = \mathcal{O}(1/t^\beta)$ and $M_t^2 = \mathcal{O}(t^{2\mu})$ for some $\beta, \mu \geq 0$, we have:*

$$\mathbb{E}[\mathrm{Reg}(T)] = \mathcal{O}(\phi(T^{-n\kappa})T^\rho + T^{1-\kappa} + T^{1-\beta} + T^{1+2\mu-\rho}) \quad \text{for all } \kappa \geq 0. \tag{17}$$

*Proof.* To simplify the proof, we will make the normalizing assumption $\theta(0) = 0$; if this is not the case, $\theta$ can always be shifted by $\theta(0)$ for this condition to hold. [Note that Examples 4 and 5 both satisfy this convention.]

With this in mind, let $\mathcal{C}$ be a convex neighborhood of $x$ in $\mathcal{K}$, and let $\mathrm{unif}_{\mathcal{C}} = \lambda(\mathcal{C})^{-1}\mathbb{1}_{\mathcal{C}}$ denote the (simple) strategy that assigns uniform probability to the elements of $\mathcal{C}$ and zero to all other points in $\mathcal{K}$. We then have:

$$h(\mathrm{unif}_{\mathcal{C}}) = \int_{\mathcal{K}} \theta(\mathrm{unif}_{\mathcal{C}}) = \int_{\mathcal{K}} \theta(\mathbb{1}_{\mathcal{C}}/\lambda(\mathcal{C})) = \int_{\mathcal{C}} \theta(1/\lambda(\mathcal{C})) = \lambda(\mathcal{C})\theta(1/\lambda(\mathcal{C})) = \phi(\lambda(\mathcal{C})).$$
(C.12)

Moreover, since $h$ is decomposable and the probability constraint $\int_{\mathcal{K}} \chi = 1$ is symmetric, the minimum of $h$ over $\mathcal{X}$ will be attained at the uniform strategy $\mathrm{unif}_{\mathcal{K}} = \lambda(\mathcal{K})^{-1}\mathbb{1}_{\mathcal{K}}$. Thus, with $\mathcal{X}$ weakly dense in $\mathrm{dom}\, h$, we obtain

$$\min h = h(\mathrm{unif}_{\mathcal{K}}) = \int_{\mathcal{K}} \theta(\mathbb{1}_{\mathcal{K}}/\lambda(\mathcal{K})) = \phi(\lambda(\mathcal{K})).$$
(C.13)

In view of all this, Corollary 3 applied to $\chi = \mathrm{unif}_{\mathcal{C}}$ yields

$$\mathbb{E}[\mathrm{Reg}_{\chi}(T)] \leq \frac{\phi(\lambda(\mathcal{C})) - \phi(\lambda(\mathcal{K}))}{\eta_{T+1}} + 2\sum_{t=1}^{T} B_t + \frac{\alpha^2}{2K}\sum_{t=1}^{T}\eta_t M_t^2,$$
(C.14)

where we used the fact that $\|\cdot\|_{\mathrm{TV}} \leq \alpha\|\cdot\|$ so $\|\cdot\|_* \leq \alpha\|\cdot\|_\infty$. The bound (16) then follows by combining the above with Lemma 1.

Regarding the bound (17), we first note that this is not a pseudo-regret bound but a bona fide bound for the learner's *expected regret* (so we cannot simply derive our point-dependent bound over $x \in \mathcal{K}$). In light of this, our first step will be to consider a "uniform" simple approximant for every $x \in \mathcal{K}$. To that end, building on an idea by Blum & Kalai [12] and Krichene et al. [38], fix a shrinkage factor $\delta > 0$ and let $\mathcal{K}_\delta(x) = \{x + \delta(x' - x) : x' \in \mathcal{K}\} \subseteq \mathcal{K}$ denote the homothetic transformation that shrinks $\mathcal{K}$ to a fraction $\delta$ of its original size and then transports it to $x \in \mathcal{K}$. By construction, we have $x \in \mathcal{K}_\delta(x) \subseteq \mathcal{K}$ and, moreover, $\mathrm{diam}(\mathcal{K}_\delta(x)) = \delta\,\mathrm{diam}(\mathcal{K})$ and $\lambda(\mathcal{K}_\delta(x)) = \delta^n\lambda(\mathcal{K})$. Then, letting $\mu_x := \mathrm{unif}_{\mathcal{K}_\delta(x)}$ denote the uniform strategy supported on $\mathcal{K}_\delta(x)$, we get

$$\mathbb{E}[\mathrm{Reg}(T)] = \mathbb{E}\left[\max_{x \in \mathcal{K}} \mathrm{Reg}_x(T)\right] \leq \mathbb{E}\left[\max_{x \in \mathcal{K}} \mathrm{Reg}_{\mu_x}(T)\right] + \delta L\,\mathrm{diam}(\mathcal{K})T,$$
(C.15)

where, in the last step, we used Lemma 1.

Now, by Proposition 1, we have

$$\mathrm{Reg}_{\mu_x}(T) \leq \frac{h(\mu_x) - \min h}{\eta_{T+1}} + \sum_{t=1}^{T}\langle e_t, p_t - \mu_x\rangle + \frac{1}{2K}\sum_{t=1}^{T}\eta_t\|\hat{u}_t\|_*^2$$

$$\leq \frac{\phi(\delta^n\lambda(\mathcal{K})) - \phi(\lambda(\mathcal{K}))}{\eta_{T+1}} + \sum_{t=1}^{T}\langle e_t, p_t - \mu_x\rangle + \frac{\alpha^2}{2K}\sum_{t=1}^{T}\eta_t\|\hat{u}_t\|_\infty^2.$$
(C.16)

and hence

$$\mathbb{E}\left[\max_{x \in \mathcal{K}} \mathrm{Reg}_{\mu_x}(T)\right] \leq \frac{\phi(\delta^n\lambda(\mathcal{K})) - \phi(\lambda(\mathcal{K}))}{\eta_{T+1}} + \mathbb{E}\left[\max_{x \in \mathcal{K}}\sum_{t=1}^{T}\langle e_t, p_t - \mu_x\rangle\right] + \frac{\alpha^2}{2K}\sum_{t=1}^{T}\eta_t M_t^2.$$
(C.17)

Thus, to proceed, it suffices to bound the second term of the above expression.

To do so, introduce the auxiliary process

$$\tilde{y}_{t+1} = \tilde{y}_t - z_t, \quad \tilde{p}_{t+1} = Q(\eta_{t+1}\tilde{y}_{t+1}),$$
(C.18)

with $\tilde{p}_1 = p_1$. We then have

$$\sum_{t=1}^{T}\langle e_t, p_t - \mu_x\rangle = \sum_{t=1}^{T}\langle e_t, (p_t - \tilde{p}_t) + (\tilde{p}_t - \mu_x)\rangle$$

$$= \sum_{t=1}^{T}\langle e_t, p_t - \tilde{p}_t\rangle + \sum_{t=1}^{T}\langle b_t, \tilde{p}_t - \mu_x\rangle + \sum_{t=1}^{T}\langle z_t, \tilde{p}_t - \mu_x\rangle$$
(C.19)

so it suffices to derive a bound for each of these terms. This can be done as follows:

1. The first term of (C.19) does not depend on $x$, so we have

$$\mathbb{E}\left[\max_{x\in\mathcal{K}}\sum_{t=1}^{T}\langle e_t, p_t - \tilde{p}_t\rangle\right] = \sum_{t=1}^{T}\mathbb{E}[\mathbb{E}[\langle e_t, p_t - \tilde{p}_t\rangle \mid \mathcal{F}_t]] \leq 2\sum_{t=1}^{T}B_t \qquad (C.20)$$

where, in the last step, we used the definition (7a) of $B_t$ and the bound

$$\langle b_t, p_t - \tilde{p}_t\rangle \leq \|p_t - \tilde{p}_t\|_1\|b_t\|_\infty \leq 2B_t. \qquad (C.21)$$

2. The second term of (C.19) can be similarly bounded as

$$\mathbb{E}\left[\max_{x\in\mathcal{K}}\sum_{t=1}^{T}\langle b_t, \tilde{p}_t - \mu_x\rangle\right] \leq \sum_{t=1}^{T}\mathbb{E}[\|\tilde{p}_t - \mu_x\|_1\|b_t\|_*] \leq 2\sum_{t=1}^{T}B_t. \qquad (C.22)$$

3. The third term is more challenging; the main idea will be to apply Proposition 1 on the sequence $-z_t$, $t = 1, 2, \ldots$, viewed itself as a sequence of virtual payoff functions. Doing just that, we get:

$$\sum_{t=1}^{T}\langle z_t, \tilde{p}_t - \mu_x\rangle \leq \frac{h(\mu_x) - \min h}{\eta_{T+1}} + \frac{1}{2K}\sum_{t=1}^{T}\eta_t\|z_t\|_*^2$$

$$\leq \frac{\phi(\delta^n\lambda(\mathcal{K})) - \phi(\lambda(\mathcal{K}))}{\eta_{T+1}} + \frac{\alpha^2}{2K}\sum_{t=1}^{T}\eta_t\|z_t\|_\infty^2. \qquad (C.23)$$

Thus, after maximizing and taking expectations, we obtain

$$\mathbb{E}\left[\max_{x\in\mathcal{K}}\langle z_t, \tilde{p}_t - \mu_x\rangle\right] \leq \frac{\phi(\delta^n\lambda(\mathcal{K})) - \phi(\lambda(\mathcal{K}))}{\eta_{T+1}} + \frac{\alpha^2}{2K}\sum_{t=1}^{T}\eta_t\sigma_t^2. \qquad (C.24)$$

Therefore, plugging Eqs. (C.20), (C.22) and (C.24) into (C.19) and substituting the result to (C.17), we finally get

$$\mathbb{E}\left[\max_{x\in\mathcal{K}}\text{Reg}_{\mu_x}(T)\right] \leq 2\frac{\phi(\delta^n\lambda(\mathcal{K})) - \phi(\lambda(\mathcal{K}))}{\eta_{T+1}} + 4\sum_{t=1}^{T}B_t + \frac{\alpha^2}{2K}\sum_{t=1}^{T}\eta_t(M_t^2 + \sigma_t^2). \qquad (C.25)$$

The guarantee (17) then follows by taking $\delta^n\lambda(\mathcal{K}) = T^{-n\kappa}$ for some $\kappa \geq 0$ and plugging everything back in (C.15). $\qquad\square$

**C.3. Dynamic regret guarantees.** We now turn to the algorithm's dynamic regret guarantees, as encoded by Theorem 2 (stated below for convenience):

**Theorem 2.** *Suppose that the Hedge variant of Alg. 1 is run with learning rate $\eta_t \propto 1/t^\rho$ and inexact models with $B_t = \mathcal{O}(1/t^\beta)$ and $M_t^2 = \mathcal{O}(t^{2\mu})$ for some $\beta, \mu \geq 0$. Then:*

$$\mathbb{E}[\text{DynReg}(T)] = \mathcal{O}(T^{1+2\mu-\rho} + T^{1-\beta} + T^{2\rho-2\mu}V_T). \qquad (20)$$

*In particular, if $V_T = \mathcal{O}(T^\nu)$ for some $\nu < 1$ and the learner's feedback is unbiased and bounded in mean square (i.e., $B_t = 0$ and $\sup_t M_t < \infty$), the choice $\rho = (1-\nu)/3$ guarantees*

$$\mathbb{E}[\text{DynReg}(T)] = \mathcal{O}(T^{\frac{2+\nu}{3}}). \qquad (21)$$

*Proof of Theorem 2.* As we discussed in the main body of our paper, our proof strategy will be to decompose the horizon of play into $m$ virtual segments, estimate the learner's regret over each segment, and then compare the learner's regret *per-segment* to the corresponding dynamic regret over said segment. We stress here again that this partition is only made for the sake of the analysis, and does not involve restarting the algorithm – e.g., as in Besbes et al. [11].

To make this precise, we first partition the interval $\mathcal{T} = [1 \mathinner{.\,.} T]$ into $m$ contiguous segments $\mathcal{T}_k$, $k = 1, \ldots, m$, each of length $\Delta$ (except possibly the $m$-th one, which might be smaller). More explicitly, take the window length to be of the form $\Delta = \lceil T^\gamma \rceil$ for some constant $\gamma \in [0, 1]$ to be

determined later. In this way, the number of windows is $m = \lceil T/\Delta \rceil = \Theta(T^{1-\gamma})$ and the $k$-th window will be of the form $\mathcal{T}_k = [(k-1)\Delta + 1 .. k\Delta]$ for all $k = 1, \dots, m-1$ (the value $k = m$ is excluded as the $m$-th window might be smaller). For concision, we will denote the learner's static regret over the $k$-th window as $\mathrm{Reg}(\mathcal{T}_k) = \max_{x \in \mathcal{K}} \sum_{t \in \mathcal{T}_k} \langle u_t, \delta_x - p_t \rangle$ (and likewise for its dynamic counterpart).

To proceed, let $\mathcal{S} \subseteq \mathcal{T}$ be a sub-interval of $\mathcal{T}$ and write $x_{\mathcal{S}}^* \in \arg\max_{x \in \mathcal{K}} \sum_{s \in \mathcal{S}} u_s(x)$ for any action that is optimal on average over the interval $\mathcal{S}$. To ease notation, we also write $x_t^* \equiv x_{\{t\}}^* \in \arg\max_{x \in \mathcal{K}} u_t(x)$ for any action that is optimal at time $t$, and $x_k^* \equiv x_{\mathcal{T}_k}^*$ for any action that is optimal on average over the $k$-th window. Then, for all $t \in \Delta_k$, $k = 1, 2, \dots, m$, we have

$$\langle u_t, \delta_{x_t^*} - p_t \rangle = \langle u_t, \delta_{x_k^*} - p_t \rangle + [u_t(x_t^*) - u_t(x_k^*)] \tag{C.26}$$

so the learner's dynamic regret over $\mathcal{T}_k$ can be bounded as

$$\mathrm{DynReg}(\mathcal{T}_k) = \sum_{t \in \mathcal{T}_k} \langle u_t, \delta_{x_k^*} - p_t \rangle + \sum_{t \in \mathcal{T}_k} [u_t(x_t^*) - u_t(x_k^*)] = \mathrm{Reg}(\mathcal{T}_k) + \sum_{t \in \mathcal{T}_k} [u_t(x_t^*) - u_t(x_k^*)]. \tag{C.27}$$

Following a batch-comparison technique originally due to Besbes et al. [11], let $\tau_k = \min \mathcal{T}_k$ denote the beginning of the $k$-th window, and let $x_{\tau_k}^*$ denote a maximizer of the first payoff function encountered in the window $\mathcal{T}_k$ (this choice could of course be arbitrary). Thus, given that $x_k^*$ maximizes the per-window aggregate $\sum_{t \in \mathcal{T}_k} u_t(x)$, we obtain:

$$\sum_{t \in \mathcal{T}_k} [u_t(x_t^*) - u_t(x_k^*)] \le \sum_{t \in \mathcal{T}_k} [u_t(x_t^*) - u_t(x_{\tau_k}^*)]$$
$$\le |\mathcal{T}_k| \max_{t \in \mathcal{T}_k} [u_t(x_t^*) - u_t(x_{\tau_k}^*)] \le 2\Delta V_k, \tag{C.28}$$

where we let $V_k = \sum_{t \in \mathcal{T}_k} \|u_{t+1} - u_t\|_\infty$. In turn, combining (C.28) with (C.27), we get:

$$\mathrm{DynReg}(\mathcal{T}_k) \le \mathrm{Reg}(\mathcal{T}_k) + 2\Delta V_k, \tag{C.29}$$

and hence, after summing over all windows:

$$\mathrm{DynReg}(T) \le \sum_{k=1}^{m} \mathrm{Reg}(\mathcal{T}_k) + 2\Delta V_T. \tag{C.30}$$

Now Theorem 1 applied to the Hedge variant of Alg. 1 readily yields

$$\mathbb{E}[\mathrm{Reg}(\mathcal{T}_k)] = \mathcal{O}\left( (k\Delta)^\rho + \Delta^{1-\kappa} + \sum_{t \in \mathcal{T}_k} t^{-\beta} + \sum_{t \in \mathcal{T}_k} t^{1+2\mu-\rho} \right) \tag{C.31}$$

so, after summing over all windows, we have

$$\sum_{k=1}^{m} \mathbb{E}[\mathrm{Reg}(\mathcal{T}_k)] = \mathcal{O}\left( \Delta^\rho \sum_{k=1}^{m} k^\rho + m\Delta^{1-\kappa} + \sum_{t=1}^{T} t^{-\beta} + \sum_{t=1}^{T} t^{2\mu-\rho} \right)$$
$$= \mathcal{O}\left( \Delta^\rho m^{1+\rho} + m\Delta^{1-\kappa} + T^{1-\beta} + T^{1+2\mu-\rho} \right). \tag{C.32}$$

Since $\Delta = \mathcal{O}(T^\gamma)$ and $m = \mathcal{O}(T/\Delta) = \mathcal{O}(T^{1-\gamma})$, we get

$$\Delta^\rho m^{1+\rho} = \mathcal{O}((m\Delta)^\rho m) = \mathcal{O}(T^{\gamma\rho} T^{(1-\gamma)(1+\rho)}) = \mathcal{O}(T^{1+\rho-\gamma}) \tag{C.33}$$

and, likewise

$$m\Delta^{1-\kappa} = \mathcal{O}(T\Delta^{-\kappa}) = \mathcal{O}(T T^{-\gamma\kappa}) = \mathcal{O}(T^{1-\gamma\kappa}). \tag{C.34}$$

Then, substituting in (C.32) and (C.30), we finally get the dynamic regret bound

$$\mathbb{E}[\mathrm{DynReg}(T)] = \mathcal{O}\left( T^{1+\rho-\gamma} + T^{1-\gamma\kappa} + T^{1-\beta} + T^{1+2\mu-\rho} + T^\gamma V_T \right). \tag{C.35}$$

To balance the above expression, we take $\gamma = 2\rho - 2\mu$ for the window size exponent (which calibrates the first and fourth terms in the sum above) and $\kappa = \beta/\gamma = \beta/(2\rho - 2\mu)$ (for the second and the third). In this way, we finally obtain

$$\mathbb{E}[\mathrm{DynReg}(T)] = \mathcal{O}\left( T^{1-\beta} + T^{1+2\mu-\rho} + T^{2\rho-2\mu} V_T \right) \tag{C.36}$$

and our proof is complete. $\qquad\square$

# D    Derivations for the bandit framework

In this appendix, we aim at deriving guarantees for the Hedge variant of Alg. 2 using template bounds from Appendix C. We start by stating preliminary results that are used in the sequel.

**D.1. Preliminary results.**    We first present a technical bound for the convex conjugate of the entropic regularizer (more on this below):

**Lemma D.1.** *For all $\psi, v \in \mathcal{V}^*$, there exists $\xi \in [0, 1]$ such that:*

$$\log \left( \int_{\mathcal{K}} \exp(\psi + v) \right) \leq \log \left( \int_{\mathcal{K}} \exp(\psi) \right) + \langle v, \Lambda(\psi) \rangle + \frac{1}{2} \langle v^2, \Lambda(\psi + \xi v) \rangle. \tag{D.1}$$

*Proof.* Consider the function $\phi \colon [0, 1] \to \mathbb{R}$ with $\phi(t) = \log \left( \int_{\mathcal{K}} \exp(\psi + tv) \right)$. By construction, $\phi(0) = \log \left( \int_{\mathcal{K}} \exp(\psi) \right)$ and $\phi(1) = \log \left( \int_{\mathcal{K}} \exp(\psi + v) \right)$. Thus, by a second-order Taylor expansion with Lagrange remainder, we have:

$$\phi(1) = \phi(0) + \phi'(0) + \frac{1}{2} \phi''(\xi) \tag{D.2}$$

for some $\xi \in [0, 1]$.

Now, for all $t \in [0, 1]$, $\phi'(t) = \frac{\int_{\mathcal{K}} v \exp(\psi + tv)}{\int_{\mathcal{K}} \exp(\psi + tv)}$, which in turns gives

$$\phi'(0) = \frac{\int_{\mathcal{K}} v \exp(\psi)}{\int_{\mathcal{K}} \exp(\psi)} = \langle v, \Lambda(\psi) \rangle. \tag{D.3}$$

As for the second order derivative of $\phi$, we have for all $t \in [0, 1]$:

$$\begin{aligned} \phi''(t) &= \frac{\partial}{\partial t} \left[ \frac{\int_{\mathcal{K}} v \exp(\psi + tv)}{\int_{\mathcal{K}} \exp(\psi + tv)} \right] \\ &= \frac{\int_{\mathcal{K}} v^2 \exp(\psi + tv) \int_{\mathcal{K}} \exp(\psi + tv) - \left( \int_{\mathcal{K}} v \exp(\psi + tv) \right)^2}{\left( \int_{\mathcal{K}} \exp(\psi + tv) \right)^2} \\ &\leq \frac{\int_{\mathcal{K}} v^2 \exp(\psi + tv) \int_{\mathcal{K}} \exp(\psi + tv)}{\left( \int_{\mathcal{K}} \exp(\psi + tv) \right)^2} = \frac{\int_{\mathcal{K}} v^2 \exp(\psi + tv)}{\int_{\mathcal{K}} \exp(\psi + tv)} \end{aligned} \tag{D.4}$$

Thus, for all $t \in [0, 1]$, we get

$$\phi''(t) \leq \langle v^2, \Lambda(\psi + tv) \rangle. \tag{D.5}$$

Our claim then follows by injecting (D.3) and (D.5) into (D.2).    □

In the next lemma, we now present an expression of the Fenchel coupling in the specific case of the negentropy regularizer $h(p) = \int_{\mathcal{K}} p \log p$.

**Lemma D.2.** *In the case of the negentropy regularizer $h(p) = \int_{\mathcal{K}} p \log p$, the Fenchel coupling for all $\psi \in \mathcal{V}^*$ and $p \in \mathrm{dom}\, h$ is given by*

$$F(p, \psi) = \int_{\mathcal{K}} p \log p + \log \left( \int_{\mathcal{K}} \exp(\psi) \right) - \langle \psi, p \rangle. \tag{D.6}$$

*Proof.* We remind the general expression of the Fenchel coupling given in (B.6):

$$F(p, \psi) = h(p) + h^*(\psi) - \langle \psi, p \rangle \quad \text{for all } p \in \mathrm{dom}\, h, \psi \in \mathcal{V}^*, \tag{D.7}$$

where $h^*(\psi) = \max_{p \in \mathcal{V}} \{ \langle \psi, p \rangle - h(p) \}$. In the case of the negentropy regularizer $h(p) = \int_{\mathcal{K}} p \log p$, we have that $\arg\max_{p \in \mathcal{V}} \{ \langle \psi, p \rangle - h(p) \} = \Lambda(\psi)$ and

$$h^*(\psi) = \langle \psi, \Lambda(\psi) \rangle - h(\Lambda(\psi)). \tag{D.8}$$

Combining the above, we then get:

$$
\begin{aligned}
h^*(\psi) &= \int_{\mathcal{K}} \psi \, \Lambda(\psi) - \int_{\mathcal{K}} \Lambda(\psi) \log \Lambda(\psi) \\
&= \int_{\mathcal{K}} \psi \, \Lambda(\psi) - \int_{\mathcal{K}} \Lambda(\psi)\psi + \int_{\mathcal{K}} \log \left( \int_{\mathcal{K}} \exp(\psi) \right) \Lambda(\psi) \\
&= \log \left( \int_{\mathcal{K}} \exp(\psi) \right).
\end{aligned}
$$

which, combined with (B.6), delivers (D.6). $\qquad\square$

Finally we state a result enabling to control the difference between the regret $\mathrm{Reg}(T)$ and $\widetilde{\mathrm{Reg}}(T)$ induced respectively by two policies $p_t$ and $\tilde{p}_t$ against the same rewards and models.

**Lemma D.3.** *For $t = 1, \ldots, T$, let $p_t, \tilde{p}_t$ be two policies with respective regret $\mathrm{Reg}(T)$ and $\widetilde{\mathrm{Reg}}(T)$ against a given sequence of models $(\hat{u}_t)_t$ for the rewards $(u_t)_t$. Then:*

$$
\mathrm{Reg}(T) \leq \widetilde{\mathrm{Reg}}(T) + \sum_{t=1}^{T} \|p_t - \tilde{p}_t\|_\infty. \tag{D.9}
$$

*Proof.* See Slivkins [47, Chap. 6]. $\qquad\square$

**D.2. Hedge-specific bounds.** We are now ready to adapt the template bound of Lemma C.1 to the Hedge case.

**Lemma D.4.** *Assuming the regularizer $h$ is the negentropy $h(p) = \int_{\mathcal{K}} p \log p$, and that the mirror map $Q$ corresponds to the logit operator $\Lambda$, there exists $\xi \in [0,1]$ such that, for all $\chi \in \mathcal{X}$ the policy (DA) enjoys the bound:*

$$
E_{t+1} \leq E_t + \langle \hat{u}_t, p_t - \chi \rangle + \left( \frac{1}{\eta_{t+1}} - \frac{1}{\eta_t} \right)[h(\chi) - \min h] + \frac{\eta_t}{2} G_t(\xi)^2. \tag{D.10}
$$

*where for all $\xi \in [0,1]$, $G_t(\xi)^2 = \langle \Lambda(\eta_t y_t + \xi \eta_t \hat{u}_t), \hat{u}_t^2 \rangle$.*

*Proof.* We know from Lemma D.4 that the policy (DA) enjoys the bound:

$$
E_{t+1} \leq E_t + \langle \hat{u}_t, p_t - \chi \rangle + \left( \frac{1}{\eta_{t+1}} - \frac{1}{\eta_t} \right)[h(\chi) - \min h] + \frac{1}{\eta_t} F(p_t, \eta_t y_{t+1}). \tag{D.11}
$$

The following lemma will help us handle the Fenchel coupling term in (D.11)

**Lemma D.5.** *For a given $t$ in the policy (DA), there exists $\xi \in [0,1]$ such that the following bounds holds:*

$$
F(p_t, \eta_t y_{t+1}) \leq \frac{\eta_t^2}{2} G_t(\xi)^2. \tag{D.12}
$$

Injecting the result given in Lemma D.5 in Eq. (D.11) yields the stated claim. $\qquad\square$

Moving forward, we are only left to prove Lemma D.5.

*Proof.* Since we are in the case of the negentropy regularizer, Lemma D.2 enables to rewrite the Fenchel coupling term of (D.11) as:

$$
F(p_t, \eta_t y_{t+1}) = \int_{\mathcal{K}} p_t \log p_t + \log \left( \int_{\mathcal{K}} \exp(\eta_t y_{t+1}) \right) - \langle \eta_t y_{t+1}, p_t \rangle. \tag{D.13}
$$

Injecting $y_{t+1} = y_t + \hat{u}_t$ in (D.13) yields:

$$F(p_t, \eta_t y_{t+1}) = \int_{\mathcal{K}} p_t \log p_t + \log\left(\int_{\mathcal{K}} \exp(\eta_t y_t + \eta_t \hat{u}_t)\right) - \langle \eta_t y_t, p_t \rangle - \langle \eta_t \hat{u}_t, p_t \rangle$$

$$= F(p_t, \eta_t y_t) + \left(\log \int_{\mathcal{K}} \exp(\eta_t y_t + \eta_t \hat{u}_t) - \log \int_{\mathcal{K}} \exp(\eta_t y_t)\right) - \langle \eta_t \hat{u}_t, p_t \rangle$$

$$= \log\left(\int_{\mathcal{K}} \exp(\eta_t y_t + \eta_t \hat{u}_t)\right) - \log\left(\int_{\mathcal{K}} \exp(\eta_t y_t)\right) - \langle \eta_t \hat{u}_t, p_t \rangle \qquad \text{(D.14)}$$

where we used the fact that $F(p_t, \eta_t y_t) = 0$.

Now, by Lemma D.1 applied to $\psi \leftarrow \eta_t y_t$ and $v \leftarrow \eta_t \hat{u}_t$, there exists $\xi \in [0, 1]$ such that

$$\log\left(\int_{\mathcal{K}} \exp(\eta_t y_t + \eta_t \hat{u}_t)\right) \leq \log\left(\int_{\mathcal{K}} \exp(\eta_t y_t)\right) + \eta_t \langle \hat{u}_t, \Lambda(\eta_t y_t) \rangle + \frac{\eta_t^2}{2} \langle \hat{u}_t^2, \Lambda(\eta_t y_t + \xi \eta_t \hat{u}_t) \rangle,$$
(D.15)

where we used the fact that $p_t = \Lambda(\eta_t y_t)$. Our claim then follows by injecting (D.15) into our prior expression for the Fenchel coupling $F(p_t, \eta_t y_{t+1})$ in the case of the Hedge variant. $\qquad\square$

**Proposition D.1.** *If we run the Hedge variant of Alg. 1, there exists a sequence $\xi_t \in [0, 1]$ such that:*

$$\mathbb{E}[\text{Reg}_x(T)] \leq \frac{\log(\lambda(\mathcal{K})/\lambda(\mathcal{C}))}{\eta_{T+1}} + L \operatorname{diam}(\mathcal{C})T + 2\sum_{t=1}^{T} B_t + \frac{1}{2}\sum_{t=1}^{T} \eta_t \, \mathbb{E}[G_t(\xi_t)^2 \mid \mathcal{F}_t],$$
(D.16)

*where $\mathcal{C}$ is a convex neighborhood of $x$ in $\mathcal{K}$.*

*Proof.* This result is obtained by using the template bound given in Lemma D.4, then by proceeding exactly as in the proofs of Proposition 1 and Theorem 1. $\qquad\square$

We stress here that Proposition D.1 *does not correspond* to the Hedge instantiation Theorem 1. Indeed, the second order term $\frac{1}{2}\sum_{t=1}^{T} \eta_t \, \mathbb{E}[G_t(\xi_t)^2 \mid \mathcal{F}_t]$ builds on results that are specific to Hedge, and is a priori considerably sharper than $\frac{\alpha^2}{2K}\sum_{t=1}^{T} \eta_t M_t^2$, the second order term of Theorem 1.

**D.3. Guarantees for Alg. 2.** For clarity, we begin by reminding the specific assumptions relative to Alg. 2. In particular, we are still considering throughout a dual averaging policy (DA) with a negentropy regularizer. We additionally assume that at each round $t$, we receive a model $\hat{u}_t$ built according to the "smoothing" approach described in Section 5 where for all $t$:

$$\hat{u}_t(x) = K_t(x_t, x) \cdot u_t(x_t)/p_t(x_t) \qquad \text{(D.17)}$$

where $K_t \colon \mathcal{K} \times \mathcal{K} \to \mathbb{R}$ is a (time-varying) *smoothing kernel*, i.e., $\int_{\mathcal{K}} K_t(x, x') \, dx' = 1$ for all $x \in \mathcal{K}$. For concreteness (and sampling efficiency), we will assume that payoffs now take values in $[0, 1]$, and we will focus on simple kernels that are supported on a neighborhood $\mathcal{U}_\delta(x) = \mathbb{B}_\delta(x) \cap \mathcal{K}$ of $x$ in $\mathcal{K}$ and are constant therein, i.e., $K^\delta(x, x') = [\lambda(\mathcal{U}_\delta(x))]^{-1} \mathbb{1}\{\|x' - x\| \leq \delta\}$.
we incorporate in $p_t$ an explicit exploration term of the form $\varepsilon_t/\lambda(\mathcal{K})$.

Under these assumptions, we may now bound both the bias and variance terms in (D.16).

**Lemma D.6.** *The following inequality holds, where $L$ is a uniform Lipschitz coefficient for the reward functions $u_t$ (as described in Assumption 1)*

$$B_t \leq L\delta_t. \qquad \text{(D.18)}$$

*Moreover, there exists a constant $C_{\mathcal{K}}$ (depending only on the set $\mathcal{K}$) such that:*

$$\sup_{\xi \in [0,1]} \mathbb{E}[G_t(\xi)^2 \mid \mathcal{F}_t] \leq C_{\mathcal{K}} \delta_t^{-n} \epsilon_t^{-1}. \qquad \text{(D.19)}$$

Note that bounding the second order term of Theorem 1 under the same assumptions would have yielded a $\delta_t^{-2n}$ factor instead of $\delta_t^{-n}$, which is a strictly weaker result!

*Proof.* We first prove (D.18). Using the fact that $u_t(x) = \int_\mathcal{K} u_t(x) K_t(x_t, x) dx_t$, we obtain:

$$|\mathbb{E}[\hat{u}_t(x) - u_t(x) \,|\, \mathcal{F}_t]| = \left| \int_\mathcal{K} K_t(x_t, x) \frac{u_t(x_t)}{p_t(x_t)} p_t(x_t) dx_t - \int_\mathcal{K} u_t(x) K_t(x_t, x) dx_t \right|$$

$$= \left| \int_\mathcal{K} (u_t(x_t) - u_t(x)) K_t(x_t, x) dx_t \right|$$

$$= [\lambda(\mathcal{U}_{\delta_t}(x))]^{-1} \left| \int_\mathcal{K} \mathbb{1}\{\|x' - x\| \le \delta_t\}(u_t(x_t) - u_t(x)) dx_t \right|$$

$$\le L[\lambda(\mathcal{U}_{\delta_t}(x))]^{-1} \underbrace{\int_\mathcal{K} \mathbb{1}\{\|x' - x\| \le \delta_t\}(u_t\|x_t) - u_t(x)\|}_{\le \lambda(\mathcal{U}_{\delta_t}(x))\delta_t} \le L\delta_t. \quad \text{(D.20)}$$

This bound is uniform (does not depend on the point $x$), and thus implies the stated inequality for $B_t$.

We now turn to (D.19). To that end, let $\xi \in [0, 1]$. We will prove a uniform bound on $\mathbb{E}[G_t(\xi)^2 \,|\, \mathcal{F}_t]$. As a preliminary it is capital to note that, $\mathcal{K}$ being convex compact, there exists constants $C_\mathcal{K}^M$ and $C_\mathcal{K}^m$ such that for all $x \in \mathcal{K}$,

$$C_\mathcal{K}^m \delta_t^n \le [\lambda(\mathcal{U}_{\delta_t}(x))] \le C_\mathcal{K}^M \delta_t^n.$$

Now, using $G_t(\xi)^2 = \langle \Lambda(\eta_t y_t + \xi \eta_t \hat{u}_t), \hat{u}_t^2 \rangle$ and $\hat{u}_t(x) = K_t(x_t, x) \cdot u_t(x_t)/p_t(x_t)$, we may write:

$$\mathbb{E}\big[G_t(\xi)^2 \,\big|\, \mathcal{F}_t\big] = \mathbb{E}\left[ \int_\mathcal{K} \Lambda(\eta_t y_t + \xi \eta_t \hat{u}_t)(x) K_t(x_t, x)^2 \frac{u_t(x_t)^2}{p_t(x_t)^2} dx \,\middle|\, \mathcal{F}_t \right]$$

$$\le \int_\mathcal{K} \overbrace{\frac{u_t(x')^2}{p_t(x')^2}}^{\le 1} p_t(x') \left( \int_\mathcal{K} \Lambda(\eta_t y_t + \xi \eta_t \hat{u}_t)(x) K_t(x', x)^2 dx \right) dx'$$

$$\le \int_\mathcal{K} \underbrace{\frac{1}{p_t(x')}}_{\ge \varepsilon_t/\lambda(\mathcal{K})} \underbrace{[\lambda(\mathcal{U}_{\delta_t}(x'))]^{-2}}_{\le (C_\mathcal{K}^m)^2 \delta^{-2n}} \left( \int_\mathcal{K} \Lambda(\eta_t y_t + \xi \eta_t \hat{u}_t)(x) \, \mathbb{1}\{\|x' - x\| \le \delta_t\} dx \right) dx'$$

$$\le \frac{\lambda(\mathcal{K})}{(C_\mathcal{K}^m)^2 \delta_t^{2n} \varepsilon_t} \int_\mathcal{K} \left( \int_\mathcal{K} \Lambda(\eta_t y_t + \xi \eta_t \hat{u}_t)(x) \, \mathbb{1}\{\|x' - x\| \le \delta_t\} dx \right) dx'$$

$$= \frac{\lambda(\mathcal{K})}{(C_\mathcal{K}^m)^2 \delta_t^{2n} \varepsilon_t} \int_\mathcal{K} \left( \int_\mathcal{K} \Lambda(\eta_t y_t + \xi \eta_t \hat{u}_t)(x) \, \mathbb{1}\{\|x' - x\| \le \delta_t\} dx' \right) dx \quad \text{(Fubini)}$$

$$= \frac{\lambda(\mathcal{K})}{(C_\mathcal{K}^m)^2 \delta_t^{2n} \varepsilon_t} \int_\mathcal{K} \Lambda(\eta_t y_t + \xi \eta_t \hat{u}_t)(x) \underbrace{\left( \int_\mathcal{K} \mathbb{1}\{\|x' - x\| \le \delta_t\} dx' \right)}_{= \lambda(\mathcal{U}_{\delta_t}(x)) \le C_\mathcal{K}^M \delta_t^n \lambda(\mathcal{K})} dx$$

$$\le \frac{\lambda(\mathcal{K})}{(C_\mathcal{K}^m)^2 \delta_t^{2n} \varepsilon_t} C_\mathcal{K}^M \delta_t^n \underbrace{\left( \int_\mathcal{K} \Lambda(\eta_t y_t + \xi \eta_t \hat{u}_t)(x) dx \right)}_{=1}$$

$$= \left( \frac{\lambda(\mathcal{K}) C_\mathcal{K}^M}{(C_\mathcal{K}^m)^2} \right) \delta_t^{-n} \varepsilon_t^{-1}. \quad \text{(D.21)}$$

This bound depends only on $\mathcal{K}$, and is notably independent on $\xi \in [0, 1]$. The result (D.19) follows directly. $\qquad\square$

We are now ready to prove Proposition 2 and Eq. (24).

**Proposition 2.** *Suppose that the Hedge variant of Alg. 2 is run with learning rate $\eta_t \propto 1/t^\rho$ and smoothing/exploration schedules $\delta_t \propto 1/t^\mu$, $\varepsilon_t \propto 1/t^\beta$ respectively. Then, the learner enjoys the bound*

$$\mathbb{E}[\text{Reg}(T)] = \mathcal{O}(T^\rho + T^{1-\mu} + T^{1-\beta} + T^{1+n\mu+\beta-\rho}). \quad (23)$$

*In particular, if the algorithm is run with $\rho = (n+2)/(n+3)$ and $\mu = \beta = 1/(n+3)$, we obtain the bound $\mathbb{E}[\text{Reg}(T)] = \mathcal{O}(T^{\frac{n+2}{n+3}})$.*

*Proof.* Let us consider a slight modification of Alg. 2 in which

- The models $(\hat{u}_t)$ received by the learner are the same models than those generated by running Alg. 2,

- At each round $t$, the action $x_t$ is sampled according to $\tilde{p}_t = \Lambda(\eta_t y_t)$ (without taking into account the explicit exploration term).

The regret of this algorithm may be bounded using the Hedge template bound stated in Proposition D.1, since we are indeed considering the regret induced by Hedge against the sequence of reward models $\hat{u}_t$[4]. Then, writing $\widetilde{\mathrm{Reg}}(T)$ for the regret induced by the policy $(\tilde{p}_t)_t$, we get

$$\mathbb{E}[\widetilde{\mathrm{Reg}}_x(T)] \leq \frac{\log(\lambda(\mathcal{K})/\lambda(\mathcal{C}))}{\eta_{T+1}} + L\,\mathrm{diam}(\mathcal{C})T + 2\sum_{t=1}^{T} B_t + \frac{1}{2}\sum_{t=1}^{T} \eta_t\,\mathbb{E}[G_t(\xi_t)^2\,|\,\mathcal{F}_t]. \tag{D.22}$$

Using the bounds presented in Lemma D.6 we then get:

$$\mathbb{E}[\widetilde{\mathrm{Reg}}_x(T)] \leq \frac{\log(\lambda(\mathcal{K})/\lambda(\mathcal{C}))}{\eta_{T+1}} + L\,\mathrm{diam}(\mathcal{C})T + 2L\sum_{t=1}^{T}\delta_t + \frac{1}{2}C_{\mathcal{K}}\sum_{t=1}^{T}\eta_t\delta_t^{-n}\epsilon_t^{-1}. \tag{D.23}$$

We are however interested in guarantees for Alg. 2, in which we play with the policy $(p_t)_t$, which slightly differs from the Hedge policy $(\tilde{p}_t)_t$. To that end, Lemma D.3 enables us to bound the difference between the regrets $\mathrm{Reg}\,T$ and $\widetilde{\mathrm{Reg}}(T)$, induced by $(p_t)_t$ and $(\tilde{p}_t)_t$ respectively. Namely we can write:

$$\mathrm{Reg}\,T \leq \widetilde{\mathrm{Reg}}(T) + \sum_{t=1}^{T}\|\tilde{p}_t - p_t\|_\infty. \tag{D.24}$$

For any $t \geq 1$, $x \in \mathcal{K}$, we have

$$|\tilde{p}_t(x) - p_t(x)| = \left|\tilde{p}_t(x) - (1-\varepsilon_t)\tilde{p}_t(x) - \frac{\varepsilon_t}{\lambda(\mathcal{K})}\right| = \varepsilon_t \cdot |\tilde{p}_t - 1/\lambda(\mathcal{K})| \leq \varepsilon_t\left(1 + \frac{1}{\lambda(\mathcal{K})}\right). \tag{D.25}$$

Injecting this in (D.24) we get

$$\mathrm{Reg}\,T \leq \widetilde{\mathrm{Reg}}(T) + \left(1 + \frac{1}{\lambda(\mathcal{K})}\right)\sum_{t=1}^{T}\epsilon_t. \tag{D.26}$$

Finally, combining (D.26) with (D.23) yields:

$$\mathbb{E}[\widetilde{\mathrm{Reg}}_x(T)] \leq \frac{\log(\lambda(\mathcal{K})/\lambda(\mathcal{C}))}{\eta_{T+1}} + L\,\mathrm{diam}(\mathcal{C})T$$
$$+ 2L\sum_{t=1}^{T}\delta_t + \frac{C_{\mathcal{K}}}{2}\sum_{t=1}^{T}\eta_t\delta_t^{-n}\epsilon_t^{-1} + \left(1 + \frac{1}{\lambda(\mathcal{K})}\right)\sum_{t=1}^{T}\epsilon_t. \tag{D.27}$$

Now, using the same reasoning as in the proof of Theorem 1 with regards to the set $\mathcal{C}$, and using $\eta_t \propto 1/t^\rho$, $\delta_t \propto 1/t^\mu$ and $\varepsilon_t \propto 1/t^\beta$ straightforwardly gives:

$$\mathbb{E}[\mathrm{Reg}(T)] = \mathcal{O}(T^\rho + T^{1-\mu} + T^{1-\beta} + T^{1+n\mu+\beta-\rho}).$$

Finally, $\rho = (n+2)/(n+3)$ and $\mu = \beta = 1/(n+3)$ gives the optimal bound:

$$\mathbb{E}[\mathrm{Reg}(T)] = \mathcal{O}(T^{\frac{n+2}{n+3}}).$$

$\square$

**Proposition 3.** *Suppose that the Hedge variant of Alg. 2 is run with parameters as in Proposition 2 against a stream of loss functions with variation $V_T = \mathcal{O}(T^\nu)$. Then, the learner enjoys*

$$\mathbb{E}[\mathrm{DynReg}(T)] = \mathcal{O}(T^{1+n\mu+\beta-\rho} + T^{1-\beta} + T^{1-\mu} + T^{\nu+2\rho-n\mu-\beta}). \tag{24}$$

*In particular, if the algorithm is run with $\rho = (1-\nu)(n+2)/(n+4)$ and $\mu = \beta = (1-\nu)/(n+4)$, we obtain the optimized bound $\mathbb{E}[\mathrm{DynReg}(T)] = \mathcal{O}(T^{\frac{n+3+\nu}{n+4}})$.*

*Proof.* We use the same virtual segmentation as in the proof of Theorem 2. As a reminder, this means that we partition the interval $\mathcal{T} = [1 .. T]$ into $m$ contiguous segments $\mathcal{T}_k$, $k = 1, \ldots, m$, each of length $\Delta$ (except possibly the $m$-th one, which might be smaller). More explicitly, take the window length to be of the form $\Delta = \lceil T^\gamma \rceil$ for some constant $\gamma \in [0, 1]$ to be determined later. In this way, the number of windows is $m = \lceil T/\Delta \rceil = \Theta(T^{1-\gamma})$ and the $k$-th window will be of the form $\mathcal{T}_k = [(k-1)\Delta + 1 .. k\Delta]$ for all $k = 1, \ldots, m-1$ (the value $k = m$ is excluded as the $m$-th window might be smaller). For concision, we will denote the learner's static regret over the $k$-th window as $\mathrm{Reg}(\mathcal{T}_k) = \max_{x \in \mathcal{K}} \sum_{t \in \mathcal{T}_k} \langle u_t, \delta_x - p_t \rangle$ (and likewise for its dynamic counterpart).

Following the proof of Theorem 2 up to (C.30), we can still write in our bandit setting:

$$\mathrm{DynReg}(T) \leq \sum_{k=1}^{m} \mathrm{Reg}(\mathcal{T}_k) + 2\Delta V_T. \tag{D.28}$$

Now Proposition 2 applied to the Hedge variant of Alg. 2 readily yields

$$\mathbb{E}[\mathrm{Reg}(\mathcal{T}_k)] = \mathcal{O}\left( (k\Delta)^\rho + \sum_{t \in \mathcal{T}_k} t^{-\beta} + \sum_{t \in \mathcal{T}_k} t^{-\mu} + \sum_{t \in \mathcal{T}_k} t^{\beta + n\mu - \rho} \right) \tag{D.29}$$

so, after summing over all windows, we have

$$\sum_{k=1}^{m} \mathbb{E}[\mathrm{Reg}(\mathcal{T}_k)] = \mathcal{O}\left( \Delta^\rho \sum_{k=1}^{m} k^\rho + \sum_{t=1}^{T} t^{-\beta} + \sum_{t=1}^{T} t^{-\mu} + \sum_{t=1}^{T} t^{\beta + n\mu - \rho} \right)$$

$$= \mathcal{O}\left( \Delta^\rho m^{1+\rho} + T^{1-\beta} + T^{1-\mu} + T^{1+\beta+n\mu-\rho} \right). \tag{D.30}$$

Since $\Delta = \mathcal{O}(T^\gamma)$ and $m = \mathcal{O}(T/\Delta) = \mathcal{O}(T^{1-\gamma})$, we get

$$\Delta^\rho m^{1+\rho} = \mathcal{O}((m\Delta)^\rho m) = \mathcal{O}(T^{\gamma\rho} T^{(1-\gamma)(1+\rho)}) = \mathcal{O}(T^{1+\rho-\gamma}). \tag{D.31}$$

Then, substituting in (D.30) and (D.28), we finally get the dynamic regret bound

$$\mathbb{E}[\mathrm{DynReg}(T)] = \mathcal{O}\left( T^{1+\rho-\gamma} + T^{1-\beta} + T^{1-\mu} + T^{1+\beta+n\mu-\rho} + T^\gamma V_T \right). \tag{D.32}$$

To balance the above expression, we take $\gamma = 2\rho - \beta - n\mu$ for the window size exponent (which calibrates the first and fourth terms in the sum above). In this way, we finally obtain

$$\mathbb{E}[\mathrm{DynReg}(T)] = \mathcal{O}\left( T^{1+n\mu+\beta-\rho} + T^{1-\beta} + T^{1-\mu} + T^{2\rho-n\mu-\beta} V_T \right) \tag{D.33}$$

and our proof is complete. $\qquad\square$

# E   Numerical experiments

Our aim in this appendix is to provide some numerical illustrations of the theory presented in the rest of our paper. All numerical experiments were run on a machine with 48 CPUs (Intel(R) Xeon(R) Gold 6146 CPU @ 3.20GHz), with 2 Threads per core, and 500Go of RAM. For a simulation horizon of $T = 2 \times 10^5$, we choose a reward function $[0, 1]$ that is a linear combination of trigonometric terms with different frequencies and amplitudes, arbitrarily drawn. Because of this analytic expression, we are able to calculte the learner's best action in hindsight (or instantaneously) and plot the relevant regret curves.

For illustration purposes, we compared 2 strategies, called "Grid" and "Kernel". The "Kernel" method is as outlined in Section 5 (cf. Alg. 2) with parameters described below. The "Grid" method involves partitioning the search space into a grid of a given mesh-size (a hyperparameter of the algorithm), and then treating the problem as a finite-armed bandit; in particular, the "Grid" strategy employs the EXP3 algorithm [7] with rewards sampled at the grid points.

In Fig. 1, we plot the mean regret for both algorithms, with different hyperparameters, over $T$ iterations. The confidence intervals are represented by the shaded areas, which corresponds to the mean value of the regret modulated by the standard deviation of our sample runs of each algorithm (computed on 92 initialization seeds for sampling, kept constant across different runs for control validation).

**Figure 1: Expected average regret**, averaged on 92 realizations for each algorithm (solid line). The variance is presented (shaded area) where we add and remove the standard deviation (computed on the 92 seeds) from the mean. Finally, the theoretical regret bound is displayed (dashed line).

**Figure 2: Two slices of the mean regret**, averaged on 92 realizations for each algorithm (solid line). Whisker at 5-95% CI , boxes at 25-75% CI and median displayed with vertical bars.

The dashed line represent in the figure corresponds to the theoretical regret bound of $T^{-\frac{1}{3}}$, which is the expected regret bound of the Kernel algorithm mean regret (without explicit exploration in our case). For performance evaluation purposes, we "slice" different snapshots of the regret in Fig. 2 at iteration counts $2 \times 10^5$ and $2 \times 10^5$. In both cases, we observe a dramatic drop in variance for the Kernel algorithm relative to the Grid strategy, with a fixed number of arms uniformly cut beforehand; we also note that the performance of the Kernel method approaches the theoretical slope of $T^{-1/3}$ that characterizes the Kernel method.

By contrast, the mean regret for the Grid approach seems to converge to a finite value which indicates a much slower regret minimization rate; on the other hand, the mean regret of the Kernel method converges to 0 at the anticipated rate.