[Reviews · NeurIPS 2020]

Review 1

Summary and Contributions: - This paper considers online non-convex optimization with bounded-mean, bounded-variance stochastic observations of the loss functions. - The authors show static and dynamic regret bounds for follow-the-regularized-leader (FTRL) over mixed strategies. They discuss measure-theoretic problems that arise from specifying regularizers and actions, in infinite-dimensional spaces.

Strengths: - This problem setting is a well-motivated extension of recent investigations into non-convex regret. - The results are presented at a very high level of generality, and the authors are very careful to address measure-theoretic concerns. - The framework gives a nice clean way to show how to get sublinear regret for continuous bandit optimization, and gives a new result about bandit dynamic regret.

Weaknesses: - Foremost concern: it seems that the only natural case for this framework is the application to bandit feedback in Section 5. Some more discussion of other natural settings captured by the framework would make the paper more convincing. This is reinforced by the fact that for Theorems 1 and 2 to be non-vacuous, the biases B_t must have some polynomial decay 1/t^\beta. The paper claims to resolve an open problem by [Krichene et al. ‘15]; could the authors clarify which open question is being resolved? At a glance (if I’m reading the same version), the discussion on the bandit problem seems to be posing a different open question than what is resolved in this paper. - Especially in light of the need for the biases to decay, it’s not obvious why the general result of Theorem 1 should justify the framework in a standalone manner. The case of unbiased feedback (Corollary 1) seems to have been addressed already by [Krichene et al. ‘15]. The - The claims of improvement of Theorem 2 (dynamic regret bounds) over [Besbes et al. ‘15] need to be clearer. The claimed benefits are (a) not requiring periodic restarts, and (b) applying to non-convex problems. To (a), if the regret comparator is moving, I’m not sure why it’s bad for an algorithm to qualitatively involve forgetting about the past, especially if the regret bound turns out to be optimal. To (b), the convexity is getting bypassed by an infinite-dimensional linearization, which changes the entire computational model. These are both good remarks for comparing this work to [Besbes et al. ‘15], but it’s not clear they should count as improvements per se. - For the bandit static regret section, see below about missing references to this well-studied problem. - The bandit dynamic regret result seems to me to be new, and worth pointing out. This would require more discussion of why the results don’t follow straightforwardly from other approaches to continuous bandit optimization; see the point about missing references. - The experimental methodology in the appendix leaves many questions. What is the domain? What is the “arbitrarily drawn” choice of reward function; what is its Lipschitz constant? Figure 1 shows that discretized-EXP3 doesn’t converge on the best arm; is this due to an unfairly chosen discretization? Without details, it is not clear what conclusion to extract from the experiments. (My overall score does not hinge on this; I don’t think papers in this line of work need to be supplemented by experiments.) - Multiple clarity issues; see below.

Correctness: - The proofs appear to be correct. - See above for issues with experiments, though I stress that this is not that important.

Clarity: - I found the paper very hard to follow, and would recommend multiple clarifications, probably requiring significant rearrangement. - The definition and role of “simple strategies” is confusing until the application to bandits. It’s claimed that the finite-sum form makes these strategies amenable to computationally easy sampling (suggesting that the reader should view k as small); then, it’s claimed that the set of simple strategies is dense in the weak topology of Radon measures (which is only true, of course, if k gets to be arbitrarily large). The latter point is misleading, given how simple strategies are actually used in section 5. - More generally, the generality of the framework makes it hard to understand in what situations a computational model is applicable. In full generality, the player receives infinite-dimensional feedback \hat u_t, and maintains infinite-dimensional scores y_t. This is only discussed briefly and non-rigorously without examples, in the discussion on simple strategies: “sampling from simple strategies can be done efficiently (especially if the geometry of K is relatively simple)”. - A suggestion that might encompass all of the above: for clarity, the exposition of the framework would benefit from carefully discussing the advantages of taking such a general view vs. something like choosing a suitable \eps-net of the decision set and reducing to the finite case.

Relation to Prior Work: - There is missing discussion about the large body of existing work on continuous-armed bandits, starting with “The continuum-armed bandit problem” [Agrawal ‘95]. Other examples: “Nearly tight bounds for the continuum-armed bandit problem” [Kleinberg ‘04], “Bandits and experts in metric spaces” [Kleinberg et al. ‘19], and as the cited [Krichene ‘15] mentions: “X-armed bandits” [Bubeck et al., JMLR ‘11]. The work would be significantly more convincing if it could be demonstrated to be a much cleaner framework in which to derive or improve some well-known results in any setting in this literature. As it stands, the only clear application of the framework is this bandit setting, so the paper needs significantly more discussion on where this result fits in.

Reproducibility: No

Additional Feedback: *** post-response *** Thanks for the response. The authors have addressed some of my concerns, but not the major ones. Discussion below: Major points: - I don't feel that my point about the lack of other natural settings was adequately addressed. I did not miss lines 143-153 in my original reading of the paper. To be clear: naming a general online non-convex problem is not a sufficient technical motivation for the framework. Online clique prediction with a quadratic form cost, with polynomially convergent entry-wise estimates of a ground-truth adjacency matrix, is indeed one such problem that can be named, but the end-to-end problem setting that fits into this framework is (to my knowledge) non-standard. The citation provided in the response is a broad survey of community detection methods; rather than providing a setting like this, it seems to be used to justify the authors' claims in the response that there can be exponentially convergent estimators of an adjacency matrix. I don't find this to specifically motivate the authors' sketch of the max clique formulation. Furthermore, even if one were to take as a given that this max clique problem is relevant, this framework recommends algorithms that maintain infinite-dimensional mixed strategies, and it's not clear that these are practical. - I don't believe that this work resolves the open question of [Krichene et al. '15]. Of course, that question is stated in rather vague terms (rather than a COLT-style fully-defined theoretical open question), but I believe that in the context of that paper, it is heavily implied that the open question is asking about an efficient algorithm (which would have to bypass the exponential-sized cover). As mentioned in my original review, once the computational model is switched to one where mixed strategies L^\infty (K) are maintained, the way to implement an algorithm in that model would have to involve an \eps-cover of the decision set or something similar. - I understand that Alg. 2, implemented with a smoothing kernel that's uniform in a ball surround x_t, runs in finite time, since the strategies maintained always remain "simple" (i.e. piecewise constant density). However, line 4 of Alg. 2, where the importance sampling step involves multiplication of two density functions, can double the number of components: each component in p_t has to split into its intersection with the new ball and its complement. So, in the worst case, this would have to run in time exponential in t, unless further structure is assumed. The worst-case constructions would look like ones that force the algorithm to maintain a cover of K. In any case, in order to talk about its relationship with other papers and open problems, some discussion in the manuscript is necessary. Minor points: - Handling of unbiased stochastic perturbations in [Krichene et al. '15]. My error; thanks. I had thought that that work was able to handle this case. - The distinction of adversarial vs. continuous was not my point in requesting a literature review. The more important point is that the computational model here (in general, maintaining mixed strategies in an infinite-dimensional space) differs from the rest of this well-established line of work, and some discussion is expected. I think the contribution has potential (as I mentioned in my original review, the dynamic non-convex regret is new and worth pointing out, and this paper presents a nice clean way to derive the bandit results via bias-variance decomposition of the importance sampling estimator), but is missing a lot of discussion. This is why my overall opinion remains.


Review 2

Summary and Contributions: This paper studies online non-convex optimization, and introduces a new setting where in each round t the learner can only get a noised version of the loss function. For solving this problem, the authors propose a dual averaging algorithm based on randomized mixed-strategy, which is proved to enjoy a sublinear static regret and a meaningful dynamic regret, when the noises are appropriately bounded. They also extend their algorithm to bandit non-convex optimization setting and prove that the algorithm enjoys sublinear theoretical results under similar assumptions.

Strengths: 1. This paper introduces an interesting and well-motived online learning problem, where in each round the loss function is non-convex and only a noised version is revealed to the learner. It can be seen as a generalized case as that considered in [28], and whether we can obtain meaningful results under such conditions is posed as an open question in [28]. 2. This paper successfully proposes an algorithm for solving the problem, and provide a general regret bound which depends on the descriptors of the noise. When the feedbacks are unbiased and bounded in mean square, this theoretical result yields a sublinear regret bound. This paper is also the first to consider dynamic regret in the non-convex online setting. It’s a little bit surprising to see that we can use the same algorithm to achieve low static regret and low dynamic regret simultaneously.

Weaknesses: 1. Although the authors motivated their algorithm from real-world problems, it seems to me that it is very difficult to implement Algorithm 1 to real-world applications (since we have to solve the problem in Eq.(7)). Moreover, since the decision set K can be of any form, I think it’s not easy to sample an action x_t\in K after we get p_t. It would be better if the authors can add more discussion on this point. 2. In Theorem 1, it seems that the algorithm can only achieve meaningful results when the bias term decreases very fast with respect to t. In some other cases (such as when the bais is fixed), the regret is linear. 3. In Theorem 2, to achieve meaningful results, one has to know the value of v in advance, which is generally impossible.

Correctness: I have read the main paper and I didn’t find any significant errors.

Clarity: The paper is generally well-written. However, I find some parts are a bit difficult to follow: For example, in Section 2, essentially, I understand that the mixed strategy is a distribution over K, but I wonder whether it is necessary for the authors to define this (in Section 2.1) by using a variety of terminologies from measurement theory, which may confuse some general audiences in the field of online convex optimization (such as myself). It would be better if the authors can present this in a more reader-friendly way (like [28]) and defer some technical details to the appendix.

Relation to Prior Work: The relation to prior work is clearly discussed in general. I noticed that the setting (non-convex optimization using mixed strategies) and the algorithm (duel averaging) are closely related to those in [28]. It would be better if the authors can add more discussion on this point.

Reproducibility: Yes

Additional Feedback: [14] showed that their Hedge algorithm could achieve sublinear regret when the decision set is convex or uniformly fat. In contrast, in this paper, (in the introduction) the authors mention that the decision set K is a subset of R^n; apart from this, it seems that there are no further assumptions on K. Is it a contribution of the paper that the assumption on K is removed, or do I miss something here?


Review 3

Summary and Contributions: This paper considers online learning with non-convex loss (or non-concave reward) when the revealed model is inexact. The proposed algorithm is based on dual averaging for the reformulated regret, expressed in terms of a mixed strategy policy. Such algorithm, while easy to derive, may face issues for the infinite dimensional feedback scenario considered in this paper. The authors showed that the issue can be mitigated by introducing a regularizer. The resultant dual averaging algorithm achieves sublinear static and dynamic regrets under several settings and choices of regularizer.

Strengths: Overall this appears to be a solid contribution. This paper has tackled a challenging theoretical problem in online learning which has remained open for a couple of years (as claimed by the authors). The proof appears to be correct and it uses a few tricks exploiting the decomposability of the regularizer.

Weaknesses: On the downside, the authors may wish to include some discussions about the computational complexity for the proposed algorithm. since it is now an infinite dimensional problem. Moreover, there remains a few items that can be clarified in the main paper. See my detailed comment below.

Correctness: Yes

Clarity: Yes

Relation to Prior Work: Yes

Reproducibility: Yes

Additional Feedback: This paper considers online learning with non-convex loss (or non-concave reward) when the revealed model is inexact. The proposed algorithm is based on dual averaging for the reformulated regret, expressed in terms of a mixed strategy policy. Such algorithm, while easy to derive, may face issues for the infinite dimensional feedback scenario considered in this paper. The authors showed that the issue can be mitigated by introducing a regularizer. The resultant dual averaging algorithm achieves sublinear static and dynamic regrets under several settings and choices of regularizer. Overall this appears to be a solid contribution. This paper has tackled a challenging theoretical problem in online learning which has remained open for a couple of years (as claimed by the authors). The proof appears to be correct and it uses a few tricks exploiting the decomposability of the regularizer. On the downside, the authors may wish to include some discussions about the computational complexity for the proposed algorithm. since it is now an infinite dimensional problem. Moreover, there remains a few items that can be clarified in the main paper. I have a few comments related to clarifying the main results and the dual averaging algorithm: - From the reviewer's understanding, a key step enabling a sublinear regret analysis for the non-convex online learning setting lies on the reformulation of regret in (1), which involve employing a mixed strategy to randomize the actions, and therefore allowing for optimization over the dual space. While this enables online optimization with sublinear regrets as shown in the current paper, such reformulation also leads to an infinite dimensional optimization. Such that the mixed strategy $p_t$ is in general an infinite-dimensional object. To this regard, is it possible to employ approximations to the policy found? For example, approximating the latter by a linear function, and what will be the regret incurred. The reviewer remarks that such issue is not found in prior work such as [2], whose FTPL algorithm requires only a finite dimensional action vector at each iteration (even though that seems to involve solving a non-convex problem for each iteration). - Regarding the regret bounds proven, the reviewer wonders what is the dependence on variance $\sigma_t$ defined in (6b). The latter parameter is only mentioned in (6b) and is not used in Theorem 1/2/3, which raises the question of what role does it play in the regret analysis. From the appendix, it seems that it was used in Section C and is upper bounded together with $M_t$. In this regard, perhaps it is even possible to combine it with $M_t$ from the beginning in (6)? Another point in Theorem 1 which may cause confusion is the convex neighborhood ${\cal C}$ and its diameter $diam({\cal C})$ as mentioned in (12). Upon the reading the appendix, it appears to be an object whose diameter can be adjusted freely to suit the regret analysis. However, its appearance in (12) may get some readers puzzled as it is unclear how can one choose the neighborhood. - The reviewer is also concerned about the claim that the obtained regrets are tight, as stated in the introduction. This statement seems to be supported by the discussions after Theorem 1, where the authors stated that online optimization with dual averaging also achieves a O(sqrt{T}) regret, same as the Hedge algorithm with unbiased, bounded feedback. However, the authors may need to add further reference to support that the O(sqrt{T}) bound for online convex optimization with DA is tight. - In the bandit setting with Hedge algorithm, we observe a worse regret that increases with the dimension $n$ than in the full (inexact) information setting. Is there any intuition behind it? ==== Post Rebuttal ==== The authors have answered my concerns satisfactorily. Meanwhile, I have read the other referee's report and I agree with R1 that the open question by Krichene et al. has not been fully resolved, since the proposed algorithm is only *practical* in a few special cases. I am lowering my score to reflect on this.


Review 4

Summary and Contributions: This paper considers online learning without convexity assumptions. The paper particularly considers randomized action selection algorithms with imperfect information based on dual averaging. The main contribution is establishing tight bounds for static and dynamic regrets. The bounds are also derived in the bandit case.

Strengths: Online learning has a long history in the machine learning and NeurIPS community. Online learning has been well studied under the convex assumptions, but still unsettled if removing the convex assumptions. This paper aims to the nonconvex online learning. This paper proposes to use dual averaging with inexact models. It follows the randomized algorithm analysis and also uses the trick of adding a regularizer. It makes sense to establish tight static and dynamic bounds by employing these tools. The tight bounds for non-convex cases have significance in the direction.

Weaknesses: Though it is a theoretical paper, it is better to set up some experiments to show the implications of the proposed methods and bounds.

Correctness: This is a theoretical paper and does not present experimental results.

Clarity: Overall, the paper is well written. However, there are some switches between convex (loss) and concave reward, non-convex and con-concave, which could confuse readers at some time. If using a table to list bounds and compare to old bounds, it would be clearer.

Relation to Prior Work: The connection to previous contributions are clearly stated, though I may not be full aware of very recent progress.

Reproducibility: Yes

Additional Feedback:

[Author Response · NeurIPS 2020]

We are grateful to the reviewers for their time and comments. In the rest of this rebuttal, we address their questions in order, tagging the reviewers concerned in each as **#RX**.

**#R1: Examples of learning with inexact models.**  The reviewer is stating as a foremost concern the contention that we are not discussing settings and examples captured by our framework (other than the bandit case). This is factually incorrect: **we are providing a detailed description of such a setting in Lines 143-153,** at the end of Section 2. In this example (online clique prediction), the optimizer gets an imperfect observation $\hat{A}_t$ of the adjacency matrix $A_t$ of the underlying, time-varying graph $\mathcal{G}_t$ (a social network, a feature map, etc.), and reconstructs the actual payoff model via the Motzkin-Straus theorem. As we explain in the text, the quality of the constructed model depends linearly on the accuracy of $\hat{A}_t$ as an estimator. In applications to social networks (see for example the highly cited paper [18] by Fortunato), $\hat{A}_t$ is typically obtained by mapping out a random portion of the graph starting with its most central nodes; as a result, $\hat{A}_t$ – and hence $\hat{u}_t$ – has *exponentially* small bias. In this way, the online clique prediction problem satisfies all the requirements of the "inexact model" framework and requires its full capacity.

We will be happy to use the extra page to expand our description of the online clique example and/or add others (e.g., online path planning with variable traffic demands; see also the point on parametric models below). Since this was the reviewer's foremost concern, we hope that the above clarifies the merits of our work.

**#R1: Unbiased feedback in Krichene et al.**  The reviewer is incorrect in claiming that Krichene et al. already deal with unbiased stochastic feedback: please see Algorithms 1 and 2 of Krichene et al. where it is clearly specified that the optimizer receives **perfect, deterministic information** on the loss functions, *not* stochastic observations thereof.

**#R1#R3: Open question of Krichene et al.**  The last phrase of the published ICML paper of Krichene et al. reads:

"*One question is whether one can generalize the Hedge algorithm to such a bandit setting, so that sublinear regret can be achieved without the need to explicitly maintain a cover*".

The Hedge variant of Algorithm 2 provides exactly such a generalization, i.e., **it achieves sublinear regret with bandit feedback and without needing to explicitly maintain a covering grid.** We hope that this resolves any doubts.

**#R1#R3: Infinite-dimensional input.**  In many practical applications, constructing an inexact model *does not* require infinite-dimensional feedback. A prime example is parametric models of the form $u_t \equiv u(\cdot; \theta_t)$ where $\theta_t$ is a partially observable, time-varying vector of parameters. This makes our framework applicable to a very wide range of relevant problems **that do not require infinite-dimensional input.** This is a key feature of the "inexact models" setting.

**#R1: Benefits over Besbes et al.**  We will rephrase accordingly; at the same time, please note that the dynamic regret framework focuses by necessity on *slowly varying* loss functions, so forgetting the past could be a severe drawback.

**#R1: Missing references.**  The cited references mainly concern *stochastic* bandits, not *adversarial* problems. This is a drastically different setting with incompatible results and guarantees; we will cite these papers to explain this in detail.

**#R1: On covering nets.**  An extended discussion on covering nets had already appeared in Krichene et al., so we did not deem it relevant to repeat one here. However, we will gladly use the extra page to discuss this.

**#R2: On the problem of Eq. (7).**  Indeed, a badly chosen $\psi$ could make (7) intractable. However, $\psi$ **is chosen by the optimizer,** a fact which mitigates this problem. In particular, we provide a series of examples with closed form expressions – like Hedge (our centerpiece algorithm) and the decomposable regularizers that we discuss in Appendix A.

**#R2: On the structure of $\mathcal{K}$.**  We clarify in the beginning of the paper that $\mathcal{K}$ is assumed convex (see Lines 13-14).

**#R2: On knowing $\nu$ in Theorem 2.**  Please note that **this is a very difficult unsolved problem**, even in the context of online *linear* optimization with unbiased stochastic input. [Certain papers [15,24] have proposed online convex optimization methods that adapt to $\nu$ with *perfect, deterministic input*, but the imperfect information case remains open.]

**#R3: Approximating $p_t$.**  Yes, representing $p_t$ as a finite-dim. object is exactly why we focus on simple strategies.

**#R3: On Agarwal et al.**  Please note that Agarwal et al. also requires full knowledge of the loss functions (see (1) in Alg. 1), so its information requirements are the same as Krichene et al. – and hence heavier than an inexact model.

**#R3: On $\sigma_t$ and $M_t$.**  Good point, thanks! Indeed, since $\mathcal{K}$ is compact, there is little point in the distinction, will do.

**#R3: On the role of $\mathcal{C}$.**  Good point, we understand how this can be confusing. As the reviewer suggests, we will explain in the main text that $\mathcal{C}$ is only needed for the analysis, not the derived bounds.

**#R3: On the tightness of $\mathcal{O}(\sqrt{T})$.**  This was shown rigorously by Abernethy et al. [1], we will give an exact pointer.

**#R1#R4: On experiments.**  On R4's comment, we would like to point out that we already provide experiments in App. E; on R1's comment, we can provide a github link with all code and notebooks for running the various experiments.

**#R4: On convex vs. concave.**  We understand R4's presentation concern, we will do a sweep to minimize switches.

**#R4: Tabulation of previous bounds.**  Excellent suggestion, will do.

[Meta-Review · NeurIPS 2020]

This paper led to a lot of discussion between the reviewers and the AC. While the reviewers appreciated the technical contributions of the paper, the main concern was that computational requirements of the proposed method are not clear. A discussion about the computational requirements is also important while comparing to prior methods. Overall, presenting convincing examples where these algorithms can be instantiated and specifically discussing about the implementation of sampling oracles will make this paper significantly stronger.